



# Evaluation of large-eddy simulations forced with mesoscale model output for a multi-week period during a measurement campaign

Rieke Heinze[1,2], Christopher Moseley[1], Lennart Nils Böske[2], Shravan Muppa[3], Vera Maurer[4,5,6], Siegfried Raasch[2], and Bjorn Stevens[1]

[1]Max-Planck-Institut für Meteorologie, Hamburg, Germany
[2]Institut für Meteorologie und Klimatologie, Leibniz Universität Hannover, Hannover, Germany
[3]Universität Hohenheim, Hohenheim, Germany
[4]Karlsruhe Institut für Technologie, Karlsruhe, Germany
[5]Goethe-Universität Frankfurt am Main, Frankfurt am Main, Germany
[6]Hans-Ertel-Zentrum für Wetterforschung

*Correspondence to:* R. Heinze (rieke.heinze@mpimet.mpg.de)

**Abstract.** Large-eddy simulations (LES) of a multi-week period during the HD(CP)[2] (High-Definition Clouds and Precipitation for advancing Climate Prediction) Observational Prototype Experiment (HOPE) conducted in Germany are evaluated with respect to mean boundary layer quantities and turbulence statistics. Two LES models are used in a semi-idealized setup through forcing with mescoscale model output to account for the synoptic-scale conditions. Evaluation is performed based on the HOPE observations. The mean boundary layer characteristics like the boundary layer depth are in a principal agreement with observations. Simulating shallow-cumulus layers in agreement with the measurements poses a challenge for both LES models. Variance profiles agree satisfactorily with lidar measurements. The results depend on how the forcing data stemming from mesoscale model output is constructed. The mean boundary layer characteristics become less sensitive if the averaging domain for the forcing is large enough to filter out mesoscale fluctuations.

## 1 Introduction

Large-eddy simulation (LES) studies have usually focused on a specific atmospheric boundary layer type often with the purpose of addressing a specific theoretical question. Many early atmospheric LES initially focused on cloud-free, convective boundary layers (e.g., Deardorff, 1970b, 1972; Moeng, 1984). Later various studies additionally investigated the effects of wind shear in the convective boundary layer (e.g. Mason, 1992; Moeng and Sullivan, 1994). The role of clouds in the dynamics of the boundary layer has motivated more sophisticated LES of cloud-topped boundary layers. Stratus and stratocumulus clouds have been considered in numerous works (e.g., Deardorff, 1976, 1980; Moeng, 1986; Stevens et al., 1998) and shallow cumulus clouds have also been successfully simulated (e.g. Sommeria, 1976; Cuijpers and Duynkerke, 1993; Brown et al., 2002; Siebesma et al., 2003). Less attention has been paid to stably stratified boundary layers because their simulation requires even higher resolutions and computer resources (compared to LES of convective situations) as the stable boundary layers are usually very shallow and exhibit small turbulence intensity. Nonetheless, there are several LES studies of the stable boundary layer (e.g. Mason and Derbyshire, 1990; Saiki et al., 2000; Beare et al., 2006). There are other studies investigating the diurnal





transition between different boundary layer types (e.g., Nieuwstadt and Brost, 1986; Sorbjan, 1997; van Stratum and Stevens, 2015).

In reality, however, different types of the atmospheric boundary layer occur consecutively if longer time periods spanning weeks to months and even years are considered. LES of these longer time periods (in the following called *long-term LES*) be-
came computationally tractable through massively parallel codes and advances in computing (Schalkwijk et al., 2012, 2015). What benefits and new insights can be gained from the long-term LES approach compared to previous studies? First of all, LES models can be regarded as virtual laboratories, in which the characteristics of atmospheric micro-scale flows can be studied and understood under controlled conditions (Neggers et al., 2012). One major practical benefit from LES is the development and improvement of boundary layer parameterization schemes (e.g., Noh et al., 2003). By testing parameterization
schemes with a multitude of different boundary layer situations (including transitions), the tuning towards special atmospheric conditions, which might even not be representative, can be avoided (Neggers et al., 2012). Furthermore, realistic long-term turbulence datasets are also of great interest in other fields of study, especially those with a high practical orientation (e.g., studies concerning wind energy (Vollmer et al., 2015) or air quality and ventilation effects in urban environments).

When focusing on LES longer than several hours the importance to include synoptic-scale meteorological conditions in LES
increases. Larger-scale forcing in terms of time-varying horizontal and vertical advective tendencies as well as larger-scale pressure gradients (geostrophic wind) should be prescribed to account for the overall larger-scale conditions. The strategy to prescribe larger-scale forcing terms has been applied in various single-column and cloud-resolving modeling studies (e.g., Randall and Cripe, 1999). Even early LES case studies (e.g., Sommeria, 1976) included synoptic-scale forcing. For idealized LES case studies focusing on a specific boundary layer type, the larger-scale forcing is usually constructed based on obser-
vations from measurement campaigns (e.g., Siebesma et al., 2003; Stevens et al., 2005). Synoptic-scale forcing can also be obtained from larger-scale models (e.g., Neggers and Siebesma, 2013) or a combination of observations and models (e.g., Baas et al., 2010; vanZanten et al., 2011; Pietersen et al., 2015). Regarding long-term simulations in the semi-idealized setup, relaxation towards a reference state given by a larger-scale model or observations can be used in combination with advective forcing to prevent model drift in time (Neggers et al., 2012).

In this study LES covering almost three weeks (19 days) of the HD(CP)$^2$ (High-Definition Clouds and Precipitation for advancing Climate Prediction) Observational Prototype Experiment (HOPE) are evaluated by comparing the results with the multi-sensor HOPE dataset specifically designed with this purpose in mind. Results of a year-long LES centered at a meteorological observational supersite were presented by Schalkwijk et al. (2015). They followed a statistical approach to assess the quality of their long-term LES by comparing yearly-averaged diurnal cycles and climatologies with those from observations
and concluded that the semi-idealized approach is stable enough to simulate a whole year of varying conditions. The present study focuses on a day-to-day comparison with observations from a measurement campaign which also accounts for spatial variability by providing measurements at three different principal measurement sites. Here, we want to tackle the question if the long-term LES approach is able to deliver a realistic boundary layer representation. Within this regard, the study is one of the first approaches to allow for a direct comparison of LES to measurements for a period longer than several days.





To asses how representative and robust the results of the present study are, two strategies are followed. One strategy is to use two well-established LES models instead of just one. Applying two LES models provides a measure for the variability among the LES - comparable to assessing observations of one quantity from multiple sensors. The other strategy is to study how the results depend on details of the setup. As the long-term LES approach relies on prescribing larger-scale forcing it is

for example important to know how sensitive the LES results are with respect to details of the forcing like the calculation of the larger-scale advective tendencies from the mesoscale model or the relaxation (nudging) to the mesoscale model. Furthermore, this gives us the opportunity to assess the extent to which mesoscale variability plays a role in determining boundary layer characteristics.

This study also has some relevance for using 3D LES in the form of a *superparameterization* in large-scale (global) models

as proposed by Grabowski (2016). In this approach, an LES model is embedded in each column of the large-scale model with horizontal grid lengths in the order of 10-50 km to account for an improved representation of small-scale processes in global models. In each global model grid box, one LES runs on a separate core of a massive parallel computer and communicates with the global model by exchanging only mean profiles during the simulation. The long-term LES approach under investigation would be representative for the superparameterization of one global model grid box.

Note that the LES statistics in this semi-idealized setup with prescribed forcing, can only provide a mean over a certain representative area. The measurements on the other hand are exposed to spatial variability and a point measurement at a certain location is only a local sample. As measurements from all available HOPE-sites are used for the comparison between LES and observations, a certain degree of variability can be expected in the measurements and compared to LES. In this sense we expect that a fair comparison between observations and LES is possible and that the measurements inform what should be expected

to be seen in the representative LES.

The paper is organized as follows. Section 2 provides a description of the LES models applied and of the setup. The relative importance of the larger-scale forcing terms is also assessed. Section 3 gives an overview of the measurement campaign HOPE and of the observations used in this study. In Sect. 4 the 19-day reference simulation is analyzed. First (Sect. 4.1), the temporal evolution of key boundary layer quantities is discussed. Next (Sect. 4.2), vertical profiles of second-moment turbulent quantities

for a cloud-free and a shallow cumulus case are compared with profiles obtained from lidar. In Sect. 5, the results of various sensitivity runs are presented. Summary and conclusions are presented in Sect. 6.

## 2   Large-eddy simulations

### 2.1   Large-eddy models

Two well-established LES models PALM (PArallelized Large-eddy simulation Model 4.0, revision 1574, https://palm.muk.uni-hannover.de

Maronga et al., 2015) and the UCLA-LES (University of California, Los Angeles Large-Eddy Simulation model, Stevens et al., 2005) are used in the present study. Both finite-difference models solve the same set of implicitly filtered, incompressible, non-hydrostatic Navier-Stokes equations including the three velocity components $u$, $v$, $w$ and the perturbation pressure $p$ as well as the transport equations for liquid water potential temperature $\theta_l$, the total water specific humidity $q_t$, rain water specific hu-





midity $q_r$ and number concentration $N_r$ on a staggered, C-type (Harlow and Welch, 1965; Arakawa and Lamb, 1977) Cartesian grid. The major differences between the two models are listed below.

1. In PALM the Boussinesq-approximation (Dutton and Fichtl, 1969) is used where the reference density is constant. UCLA-LES solves the equations in the less constrained anelastic approximation (Ogura and Phillips, 1962) allowing for a varying reference density with height.

2. Sub-grid scale (SGS) turbulence closure is prognostic in PALM by solving the equation for the SGS turbulence kinetic energy according to Deardorff (1980) and diagnostic in UCLA-LES using a classical Smagorinsky (1963) scheme.

3. PALM uses a fifth-order advection scheme based on Wicker and Skamarock (2002) for both, momentum and scalars. In UCLA-LES a fourth-order central advection scheme is applied for momentum and a monotone second-order scheme with flux-limiter for scalars.

4. PALM includes a Lagrangian cloud model and was often used in studies discussing shallow-convection (e.g., Riechelmann et al., 2015; Hoffmann et al., 2015; Hoffmann, 2016). UCLA-LES incorporates a hierarchy of microphysical models and representations of radiative transfer and was applied in studies more focusing on deep-convection (e.g., Rieck et al., 2015; Schlemmer and Hohenegger, 2016).

PALM and UCLA-LES both apply the fractional-step method to ensure incompressibility of the flow and the resulting Poisson-equation for the perturbation pressure is solved by a fast Fourier transform. The cloud water specific humidity $q_c$ is obtained via the simple saturation-adjustment scheme. The warm microphysics scheme of Seifert and Beheng (2001, 2006) and Seifert (2008) is applied and Monin-Obukhov similiarity theory is used at the surface. A no-slip condition is applied to the horizontal velocity components at the surface. The horizontal boundaries are cyclic and both models use a third-order Runge-Kutta method with a variable time step to advance in time. The parallelization method follows a 2D domain-decomposition using Message Passing Interface for inter-process communication.

## 2.2 Forcing with mesoscale model output

To account for synoptic-scale forcing, the effects of larger-scale pressure gradients, horizontal advection and vertical motions have to be prescribed in LES. However, the usage of lateral periodic boundary conditions constrains how the synoptic scales can be represented. As horizontal LES domain-scale gradients cannot be represented, larger-scale advection, pressure gradients and vertical motions are assumed to be horizontally homogeneous, but they may vary in time and height. This approach has direct implications on how larger-scale phenomena can be represented in the LES, for instance frontal passages. In the presence of a front, the flow field exhibits strong local gradients perpendicular to the front. However in the LES, a front would arrive and depart from the entire domain simultaneously at a specific height due to the periodic boundary conditions. Thus, the evolution of frontal passages is represented in time rather than in space (Schalkwijk et al., 2015).

Time-dependent surface conditions, which are representative for the entire LES domain, are required. To facilitate the comparison between the two LES models, surface values are prescribed instead of using a land-surface model.



The larger-scale forcing can be generated from 3D output of a larger-scale (global or limited area) climate or numerical weather prediction model (e.g., Neggers and Siebesma, 2013). Creating larger-scale forcing solely from measurements is also possible (e.g., Grabowski et al., 1996), or a combination (blending) of larger-scale model and observations can be applied (e.g. Baas et al., 2010; Bosveld et al., 2014). Here, the forcing if calculated from analysis output of the operational mesoscale numerical weather prediction model COSMO-DE (Baldauf et al., 2011, denoted as COSMO hereinafter). The COSMO analysis is thought to provide a good estimate of a current state as it is a combination of model output and assimilated measurements. COSMO is also denoted as the host model in the following.

The larger-scale (LS) tendencies for the governing equations are calculated as follows. The effect of the larger-scale pressure gradient (LSP) enters the horizontal momentum equations

$$\left.\frac{\partial u_i}{\partial t}\right|_{\mathrm{LSP}} = \varepsilon_{i3j} f_3 u_{\mathrm{g},j}, \tag{1}$$

where $u_{\mathrm{g},i} = (u_{\mathrm{g},1}, u_{\mathrm{g},2}, 0)$ denotes the geostrophic wind vector which is calculated by means of the larger-scale pressure ($p_{\mathrm{LS}}$) gradients and density ($\rho_{\mathrm{LS}}$) as $u_{\mathrm{g},1} = -(\rho_{\mathrm{LS}} f_3)^{-1} \partial p_{\mathrm{LS}}/\partial x_2$, $u_{\mathrm{g},2} = (\rho_{\mathrm{LS}} f_3)^{-1} \partial p_{\mathrm{LS}}/\partial x_1$ and $u_{\mathrm{g},3} = 0$. Einstein summation convention for repeated indices is used, $f_i = (0, 2\Omega\cos(\phi), 2\Omega\sin(\phi))$ denotes the Coriolis parameter, where $\Omega$ is the angular speed of the Earth and $\phi$ the geographical latitude.

The contributions due to LS horizontal advection (LSA) and vertical advection (*subsidence*, SUB) enter the scalar prognostic equations only:

$$\left.\frac{\partial \varphi}{\partial t}\right|_{\mathrm{LSA}} = -\left( u_{\mathrm{LS},1}\frac{\partial \varphi_{\mathrm{LS}}}{\partial x_1} + u_{\mathrm{LS},2}\frac{\partial \varphi_{\mathrm{LS}}}{\partial x_2} \right), \tag{2}$$

$$\left.\frac{\partial \varphi}{\partial t}\right|_{\mathrm{SUB}} = -u_{\mathrm{LS},3}\frac{\partial \varphi}{\partial x_3} \quad \text{with} \quad \varphi \in \{\theta_\mathrm{l}, q_\mathrm{t}\}. \tag{3}$$

All three larger-scale velocity components $u_{\mathrm{LS},i}$ and scalar components $\varphi_{\mathrm{LS}}$ are needed. Note that the LSA contribution (2) is horizontally homogeneous whereas the SUB contribution (3) is not. Here, the horizontal homogeneous subsidence velocity $u_{\mathrm{LS},3} = w_{\mathrm{SUB}}$ is combined with the local gradient of the LES scalar $\varphi$. This ensures that the tendencies are strongest where the local scalar gradients are largest and $w_{\mathrm{SUB}}$ is not negligible (which is usually at the top of the boundary layer).

The simulations presented in this study use Newtonian relaxation (nudging) additionally to the previously discussed larger-scale components. The main function of nudging in the larger-scale forcing framework is to prevent excessive model drift in time (Neggers et al., 2012). This drift may be introduced by errors in the LES or from systematic errors in the larger-scale forcing terms. By means of nudging the simulated flow is adjusted to the flow situation of the host model (Anthes, 1974; Stauffer and Bao, 1993). This is an additional possibility to account for larger-scale processes in an LES. However, relaxation has to be handled with care, since it represents no real physical process (Randall and Cripe, 1999). To preserve turbulent structures the applied nudging tendency is horizontally homogeneous in analogue to the LSP and LSA tendencies and it is





given by

$$\left.\frac{\partial \varphi}{\partial t}\right|_{\mathrm{NUD}} = -\frac{\langle\varphi\rangle - \varphi_{\mathrm{LS}}}{\tau} \quad \text{with} \quad \varphi \in \{u, v, \theta_{\mathrm{l}}, q_{\mathrm{t}}\}, \tag{4}$$

where the angle brackets ($\langle...\rangle$) denote the horizontal average of the LES variable and $\tau$ is the relaxation time-scale which defines the strength of the nudging. With a small $\tau$, the horizontal averages of the prognostic variables are adjusted relatively

fast towards the corresponding state of the host model. A nudging time scale of $\tau = 6$ h is used which is long enough for the fast boundary layer physics to develop their own unique state and short enough, so that larger-scale disturbances, such as weather fronts, can be represented in the LES (Neggers et al., 2012; Schalkwijk et al., 2015).

The larger-scale tendency terms (1)-(4) are calculated from the operational COSMO analysis data which have a horizontal and temporal resolution of 2.8 km and 3 h, respectively. Thus, the larger-scale forcing terms used in this study do not stem from

pure model output as the analysis is composed of a combination of model output and assimilated measurements. It should be noted that the larger-scale tendencies should not contain any impacts of small scale phenomena which are explicitly resolved by the LES. Thus, the COSMO data is averaged spatially to filter out these scales. The averaging procedure is further described in Appendix A. The resulting larger-scale forcing profiles are linearly interpolated in time between every three hours to obtain a forcing at every time-step in the LES.

## 2.3   Setup

The reference simulation performed with both models (denoted as *RP* and *RU* for PALM and UCLA-LES, respectively) consists of a continuous 19-day simulation covering 24 April to 12 May 2013 over the HOPE region. An isotropic grid spacing of $\Delta = 50$ m is used up to a height of 5 km above ground. Above, vertical grid stretching is applied resulting in a model top of about 13 km. Note that due to the underlying assumption of incompressibility in the set of model equations, the results above

a height of approx. 5 km should be interpreted with care especially for PALM due to the Boussinesq approximation used. A model top of 13 km is chosen nonetheless, as then the evolution of the prognostic variables above a certain height can be almost entirely ascribed to the larger-scale and deep-convective events in the forcing may find some representation in the LES. The horizontal extension of the modeling domain is 48 km$\times$48 km. In total, the model domain is resolved by $960 \times 960 \times 144$ grid cells. Figure 1a shows the topography in a 50 km$\times$50 km domain around the central HOPE region. Apart from the Eifel

mountain range in the south west of the region, the domain is rather flat which is reflected by using a flat homogeneous surface in the LES.

As explained in Sect. 2.4 and appendix A, the larger-scale forcing data is constructed by averaging COSMO analysis data. The center of the averaging domain is located at 6.375°E and 50.875°N which is centered in the HOPE-region (see Sect. 3). The larger-scale forcing data is averaged over a domain with the size of 2.0°$\times$2.0° on the geographical grid (80$\times$80 COSMO

grid points) to eliminate small scale fluctuations. This corresponds to a zonal and meridional extension of the averaging domain of 140 km and 222 km, respectively. The latitude is set to $\phi = 50.92°$ to define the Coriolis parameter for the HOPE region. At the surface, temperature and humidity are prescribed (Dirichlet conditions). The roughness length $z_0$ for momentum is adopted from the averaged COSMO data and, thus, depends on the chosen averaging domain. It results in a value of $z_0 = 0.4493$ m





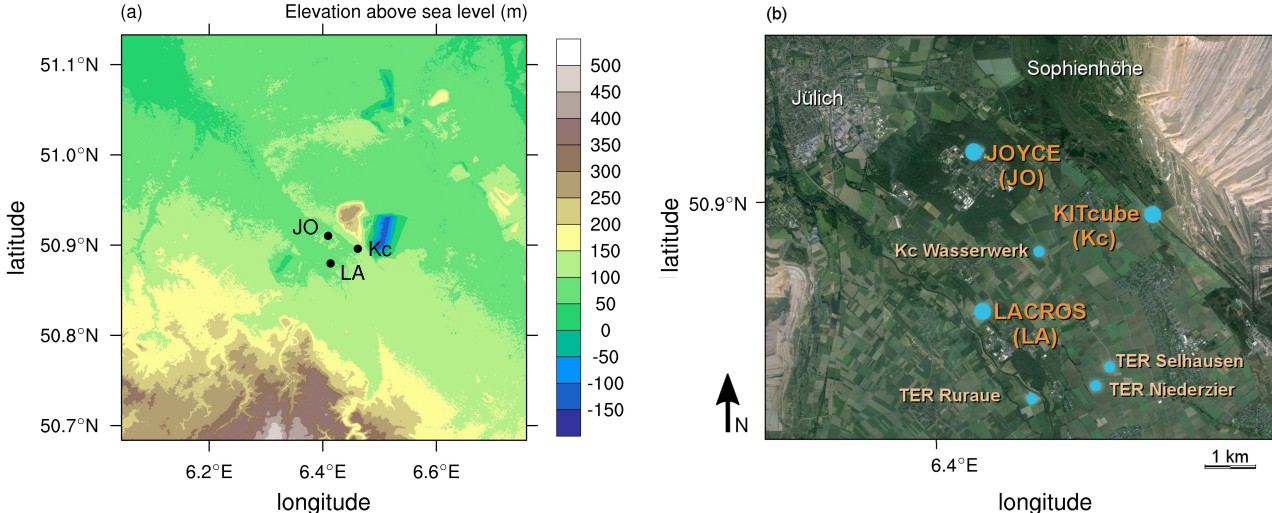

**Figure 1.** Location of different measurement sites during the HOPE campaign. The abbreviations JOYCE (JO), LACROS (LA) and KITcube (Kc) denote the three principal measurement sites Jülich ObservatorY for Cloud Evolution, Leipzig Aerosol and Cloud Remote Observations System and the Karlsruhe Institute of Technology cube, respectively. Panel a shows the topography in a 50 km × 50 km domain centered around JOYCE (source: ASTER GDEM Validation Team (2011)) and panel b provides a closer view on the HOPE measurement sites (source: Google Maps). Additional surface flux measurements were taken at the Kc site Wasserwerk (Kc Was) and the TERENO (TER) sites Ruraue (TER Rur), Selhausen (TER Sel) and Niederzier (TER Nied).

for the chosen $2.0° \times 2.0°$ averaging domain. The roughness length for scalars is usually smaller than that for momentum (Brutsaert, 1975) and chosen to be $0.1 \cdot z_0$. By constructing the forcing data-set as described, it is assumed to be representative for the HOPE area.

Note that the LES are run without radiation (neither interactive nor prescribed). Radiation is neglected as the radiative
cooling rates are usually an order of magnitude smaller than the heating rates from the surface heat flux in the mixed layer (Stull, 1988). However, through the use of nudging the effect of radiation can be regarded as indirectly accounted for.

### 2.4 Relative importance of larger-scale forcing terms

The impact of the larger-scale forcing terms on the numerical solution is evaluated and quantified. For that purpose the budget terms of the prognostic equations for liquid water potential temperature $\theta_l$ and total water specific humidity $q_t$ of the simulation
*RP* are compared. Figure 2 shows the single tendency terms which were horizontally and also vertically averaged. The vertical average is taken between the surface and the height of the boundary layer $z_i$ (in case $z_i < 500\,$m, the upper limit for the averaging is 500 m to obtain also meaningful information during night times (robust statistics) where the boundary layer is resolved by a few grid points only).

It is apparent, that during daytime the fast physics have the largest impact on the numerical solution on most of the days.
The impact of the different larger-scale forcing terms are comparably small. Sometimes (e.g., 26 April, 5 May, 11 May) the





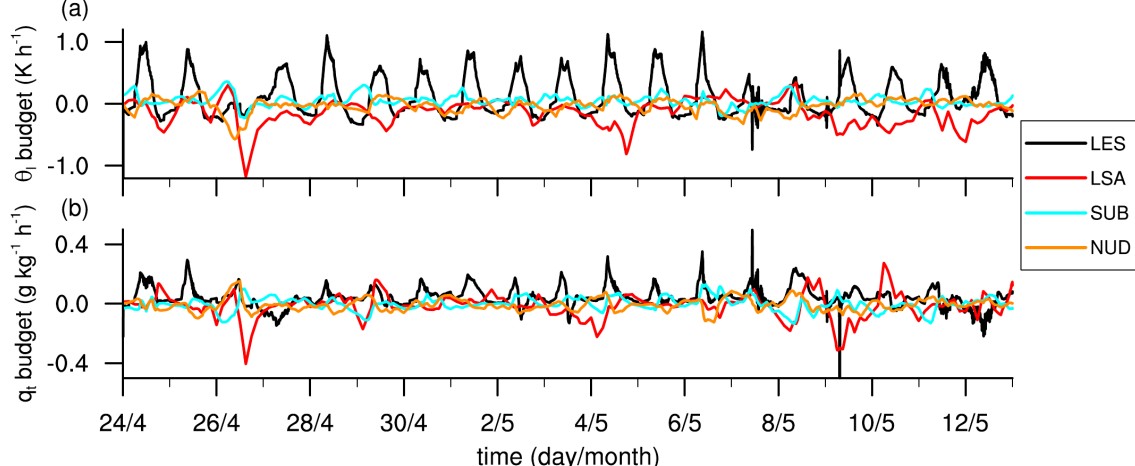

**Figure 2.** Temporal evolution of the vertically averaged budget terms of liquid water potential temperature (panel a) and total water specific humidity (panel b) of case RP. The black lines (LES) show the sum of fast LES physics (advective, subgrid diffusive and microphysical) tendencies, the red lines (LSA) denote the tendencies due to larger-scale horizontal advection, the cyan lines (SUB) show the larger-scale subsidence tendencies and the orange lines (NUD) denote the nudging tendencies. The vertical average is taken between the surface and the height of the boundary layer $z_i$ (in case $z_i < 500\,\mathrm{m}$, the upper limit for the averaging is 500 m).

larger-scale forcing terms are of opposite sign also. A clear exception is 26 April, on which a frontal passage occurs (see Sect. 3). Here, the fast physics have almost no impact on the numerical solution, and the larger-scale forcing terms dominate the change of $\theta_l$ and $q_t$ inside the boundary layer. Before noon on 26 April the LSA and SUB tendencies heat the boundary layer, and then the LSA tendencies cause a rapid and strong cooling. However, judging from the nudging tendencies for $\theta_l$, this cooling should begin some hours earlier. This circumstance may be caused by the low temporal resolution of the forcing data (3 h intervals). As the nudging tendencies are corrective tendencies, they are also a measure of the deviation between the states of COSMO and the LES. Since the nudging tendencies are generally smaller than the LSA and SUB tendencies, the latter are a sufficient representation of larger-scale physics. However, days with strong larger-scale forcing usually show slightly larger nudging tendencies.

# 3 HOPE

HOPE took place near Jülich (located in the western part of Germany) in April and May 2013. The agricultural area around the permanent observational site JOYCE (Jülich ObservatorY for Cloud Evolution (JO) at 50.907° N / 6.414° E / 111 m AMSL, Löhnert et al., 2015) was chosen to employ various in-situ and remote sensing instruments to capture a most complete set of atmospheric parameters at a high temporal and spatial resolution. JOYCE was complemented by two additional measurement sites, LACROS (Leipzig Aerosol and Cloud Remote Observations System (LA) at 50.880° N / 6.415° E / 99 m AMSL, Bühl et al., 2013) and the KITcube (Karlsruhe advanced mobile observation platform (Kc) at 50.897° N / 6.464° N / 110 m





AMSL, Kalthoff et al., 2013) during the HOPE period. The locations of the three sites and Jülich are shown in Fig. 1. Additional surface flux measurements used in this study were obtained at KITcube (Kc) site Wasserwerk and the three TERENO (TERrestrial Network of Observations (TER), Zacharias et al., 2011) sites Selhausen, Ruraue and Niederzier (see Fig. 1b), where energy balance stations were located. Within a 50 km × 50 km domain centered around JOYCE, the Eifel mountain

range is located south-west of the HOPE domain(see Fig. 1a). The most significant orographic element in the area around the HOPE sites is the Sophienhöhe (which can be seen in the upper right part of Fig. 1b) with a maximum altitude of 301 m AMSL. This element results from an open-pit mine located east of Sophienhöhe. The measurements during HOPE were taken by a multitude of instruments, such as Doppler lidars, Raman lidars, differential absorption lidar, ceilometers, microwave radiometers, cloud Doppler radars, meteorological towers, eddy-covariance stations and radiosondes. However, only a selection of these

measurements are actually used in this study, as the main emphasis is put on boundary layer characteristics and turbulence.

The 19-day period from 24 April to 12 May 2013 was chosen for the following reasons. This period contains different weather regimes (clear-sky, convective, cloudy, frontal and post-frontal situations). Furthermore, during this time-span 7 of the 18 conducted intensive observation periods (IOPs) took place in which the temporal coverage of measurements was higher (e.g., radiosondes were launched every 2 h during day time). Moreover, it covers the passage of a frontal system, i.e., an event

which is strongly controlled by fast changing larger-scale flow conditions. The frontal passage allows to study how the LES models react to such forcings. The selected period is too short for a feasible statistical analysis as conducted by Schalkwijk et al. (2015), but it is long enough to showcase and analyze the general capability to perform long-term LES.

The synoptic conditions during the 19-day period can be grouped into four different periods. During the first two days (24-25 April) high-pressure was dominating the HOPE-area resulting in a calm clear-sky (24 April) and a shallow cumulus

(25 April) day. On 26 April the situation changed noticeable as a frontal system passed from north-westerly directions over the HOPE-domain accompanied by an overcast, rainy situation and followed by three days (27-29 April) under post-frontal, overcast conditions where temperatures were significantly lower than before. The third period covering 30 April to 6 May was characterized by a calm, high-pressure period with mostly low- to mid-level convective clouds (where 3 May and 4 May were even clear-sky days). The last period began on 7 May with strong convective events (local thunderstorms). The following days

were determined by local troughs of low-pressure-systems forming over England resulting in a rough and predominantly wet period with westerly gusts up to 14 m s$^{-1}$. In terms of clouds this evolution is also apparent in Fig. 4a, which displays the Cloudnet target classifications (Illingworth et al., 2007) at LACROS site.

## 4  Reference simulation

To obtain a first visual impression of the LES data sets, snapshots of four different days (one out of each of the four weather

periods previously described) of the PALM reference simulation *RP* are compared with images from the total sky imager TSI-880 (Löhnert et al., 2015) at JOYCE site. Additionally, horizontally averaged mean profiles of potential temperature $\theta$, specific humidity $q_v$ from PALM and radiosondes launched at 11 UTC at KITcube site together with simulated cloud ($q_c$) and rain ($q_r$) water specific humidity (if present), are shown in Fig. 3. The snapshots were taken at 11 UTC on each day corresponding to the



**Table 1.** Overview of the HOPE measurements used in this study.

| Variables | Explanation | Device | Location | Time span | References |
|---|---|---|---|---|---|
| $z_i$ | boundary layer height | Doppler lidar HALO | JOYCE | 04/24-05/12 | Schween et al. (2014) |
| | | Raman lidar Polly[XT] | LACROS | 04/24-05/12 | Althausen et al. (2009) |
| | | radiosondes (Graw DFM-09) | KITcube | 04/24-05/12 | Kalthoff et al. (2013) |
| $cc$ | cloud cover | Total Sky Imager TSI-880 | JOYCE | 04/24, 04/26, 05/05, 05/10 | Löhnert et al. (2015) |
| IWV, LWP | integrated water vapor, liquid water path | micro wave radiometer HATPRO | JOYCE | 04/24-05/12 | Löhnert et al. (2015), Steinke et al. (2015) |
| shf, lhf | surface sensible heat flux, surface latent heat flux | energy balance stations | KITcube Kc Wasserwerk TER Selhausen TER Niederzier TER Ruraue | 04/24-05/12 | Kalthoff et al. (2013), Maurer et al. (2016), Graf et al. (2010), Zacharias et al. (2011) |
| $\overline{w'^2}$ | vertical velocity variance | Doppler lidar WLS7-V2 ($z < 400\,\mathrm{m}$), Doppler lidar WindTracer WTX ($z \geq 400\,\mathrm{m}$) | KITcube | 04/24, 05/05 | Maurer et al. (2016) |
| $\overline{T'^2}$ | temperature variance | rotational Raman lidar RRL | KITcube | 04/24, 05/05 | Behrendt et al. (2015) |
| $\overline{\rho_v'^2}$ | absolute humidity variance | water vapor differential absorption lidar WVDIAL | KITcube | 04/24, 05/05 | Muppa et al. (2016) |
| $z_{cb}$, $z_{ct}$, $d_c$ | cloud base height, cloud top height, cloud layer depth | Cloudnet Cloudnet ceilometer CHM15k | JOYCE LACROS JOYCE | 04/24-05/12 | Illingworth et al. (2007) Illingworth et al. (2007) Löhnert et al. (2015) |
| $T_{2m}$, $|v|_{h,120\,\mathrm{m}}$, wdir$_{120\,\mathrm{m}}$ | temperature at 2 m, wind speed at 120 m, wind direction at 120 m | 120 m meterological tower | JOYCE | 04/24-05/12 | Löhnert et al. (2015) |

Most of the data sets are available via the HD(CP)$^2$ data portal at https://icdc.zmaw.de/hdcp2.html.

launching time of the radiosondes. The visualization of simulated cloud fields, which was performed with the Visualization and Analysis Platform for Ocean, Atmosphere, and Solar Researchers (VAPOR, Clyne et al., 2007), allows for a first impression about the diversity of weather conditions encountered in the simulations.



Comparing sky imager and volume rendered cloud fields of the four days visually (left and middle columns of Fig. 3), it can be noted that the simulated cloud types agree qualitatively with the observed ones. The three-layer vertical structure in the boundary layer on 24 April is principally reproduced by PALM (Fig. 3c). However, the potential temperature is about 2 K lower than measured in the well-mixed layer and up to 1 K lower above. On 26 April, the day where the front passes the HOPE

region, a significant amount of clouds and precipitation is simulated at 11 UTC (Fig. 3f). The temperature profile of PALM is reproduced very well. However, PALM simulates a well-mixed humidity layer below 1.5 km which is not seen in the sounding. Similar to 24 April, the boundary layer and lower tropospheric layer are about 1 to 2 K colder than observed on 5 May (Fig. 3i) and also on 10 May (Fig. 3l). The vertical structure is reproduced well on both latter days.

## 4.1 Temporal evolution

### 4.1.1 Principal character of the simulated days

To provide an overview we first show how well the principal character of the day in terms of clouds and precipitation is represented in the LES over the course of the 19-day period. A qualitative comparison of cloud water and cloud rain produced by the LES with the Cloudnet product (Illingworth et al., 2007) at LACROS site complemented with the weather overview archive produced during HOPE, is presented. Note that Cloudnet is a composite measurement product, which is derived from

ceilometer, cloud radar, microwave radiometer and output from the COSMO model (e.g., Löhnert et al., 2015).

Figure 4a shows the Cloudnet target classification at LACROS. Roughly, the period consists of two clear-sky days (24 April and 4 May), nine predominantly cloudy days (25, 28, 29 and 30 April, 1, 2, 3, 5 and 6 May) and eight days where precipitation occurred (26 and 27 April, 7, 8, 9, 10, 11 and 12 May). Applying the same qualitative criteria (clear-sky, cloudy, rainy) to the PALM and UCLA-LES representation of clouds and precipitation in terms of cloud and rain water specific humidity

(Fig. 4c,d), the following summary can be given (see also Tab. 2). On two days (25 April, 1 May), both LES models were not able to simulate shallow cumuli during the day although shallow cumuli were observed. Precipitation was simulated on too few days. UCLA-LES did not simulate precipitation on three days and PALM on one day. This sums up to a qualitative agreement in the principal character of the day on 16 days for PALM and 14 days for UCLA-LES, which is an agreement of 84 % and 74 %, respectively.

Comparing specific cloud and rain water of the two LES models with the COSMO forcing (Fig. 4b) we want to stress that only a warm-rain microphysics scheme has been applied in both models. This clearly restricts the possibility to realistically form upper-level clouds and precipitation in the LES as these processes usually require the ice-phase in mid-latitudes. Nonetheless, PALM and UCLA-LES both find a representation of higher level clouds, especially on days with strong impact of larger-scale forcing like the frontal day of 26 April. The shallow cloud layers usually form on top of the boundary layer as

can be seen in Fig. 4c,d. These cloud layers usually find a good representation when using a warm-microphysics scheme only. Anyhow, the more challenging situations for PALM and UCLA-LES seem to be simulations of proper shallow cumulus layers on some days (25 April and 1 May).



**Table 2.** Summary on qualitative agreement in the principal character of the simulated days compared to Cloudnet and HOPE weather overview archive.

| Criteria | PALM days | no. days | UCLA-LES days | no. days |
|---|---|---|---|---|
| LES days without shallow cumuli, when shallow cumuli were observed | 25/04, 01/05 | 2 | 25/04, 01/05 | 2 |
| LES days without rain, when rain was observed | 08/05 | 1 | 26/04, 08/05, 09/05 | 3 |
| LES days with qualitative agreement to observations | remaining | 16 | remaining | 14 |

The cloud and precipitation structure over the 19 days is very similar in both models, but UCLA-LES produces a lesser amount of cloud and rain water (the latter leading to two more days of qualitative misrepresentation in UCLA-LES as compared to PALM, see Tab. 2). This difference roots in the usage of different advection schemes for scalars as PALM uses a fifth-order scheme whereas UCLA-LES applies a monotone second-order scheme with flux-limiter (see Sect. 2.1). Monotone schemes

show a rather diffusive character (Durran, 1999). Thus, the horizontal and vertical gradients are smoothed more strongly in UCLA-LES than in PALM which could even lead to a complete damping of small amplitudes of humidity and updrafts prohibiting formation of weak clouds and precipitation. Furthermore, the specific rain water is slightly better represented in the LES than in COSMO. COSMO shows much more rain than observed.

### 4.1.2 Boundary layer depth

The boundary layer depth $z_i$ is one of the major defining characteristics of the boundary layer. In this study, $z_i$ has to be determined for different types of boundary layers (stable, convective, cloud-topped) and the respective transitional phases, because several diurnal cycles are simulated. Therefore, a robust criterion, that works well for the different boundary layer types, has to be chosen for an adequate determination of $z_i$. However, most established methods are closely tied to one boundary layer type (e.g., the height of the minimum buoyancy flux for the convective boundary layer). Thinking of a broader definition

of the boundary layer, it can be identified as the layer in which turbulent mixing occurs due to the presence of the surface. The dimensionless Richardson number Ri is defined as the ratio of buoyancy to shear production of turbulence kinetic energy. The boundary layer depth can also be defined as the height where Ri exceeds a critical value as Ri provides a measure of the dynamic stability of the flow. Criteria based on Ri have been frequently used in a number of studies over the last decades (e.g., Richardson et al., 2013, and references therein). The bulk Richardson number $\mathrm{Ri_b}$ is derived from the gradient Richardson

number by approximating local gradients to a finite difference across a layer and it is defined as

$$\mathrm{Ri_b} = \left( \frac{g}{\theta_{v,s}} \right) \frac{\theta_v - \theta_{v,s}}{u_1^2 + u_2^2} \cdot z, \tag{5}$$

where g is the acceleration due to gravity, $\theta_v$ denotes the virtual potential temperature and $\theta_{v,s}$ its value close to the surface. Following classical theory (Taylor, 1931), turbulence of a homogeneous stably stratified sheared flow in steady state decays, if the gradient Richardson number exceeds a value of 0.25. In the definition (Eq. (5)), $\mathrm{Ri_b}$ is defined from the surface upwards.





If $z$ is replaced by the boundary layer height $z_i$, $\mathrm{Ri_b}$ becomes the critical bulk Richardson number whose value depends on stability (e.g., Richardson et al., 2013; Basu et al., 2014). However, this dependence is neglected in this study and a value of $\mathrm{Ri_{b,c}} = 0.25$ is assumed to be valid for all stability regimes. This applied value also lies in the interval for the critical bulk Richardson number $0.2 < \mathrm{Ri_{b,c}} < 0.5$, proposed by Zilitinkevich and Baklanov (2002).

In PALM and UCLA-LES, $z_i$ is determined locally (at each grid point in the horizontal domain). Starting at the lowest prognostic level and continuing upwards, $\mathrm{Ri_b}$ is calculated using Eq. (5) until $\mathrm{Ri_b} > \mathrm{Ri_{b,c}} = 0.25$. The height of the grid point at which the critical value is exceeded is then assumed to coincide with $z_i$. For $\theta_{v,s}$ the second prognostic level above the surface is used. The resulting 2D field of $z_i$ is then averaged horizontally and the horizontal variability is quantified by means of retaining the standard deviation.

Figure 5 shows the temporal evolution of the boundary layer depth. The aforementioned spatial variability is depicted as twice the standard-deviation in light gray shading for PALM and light green shading for UCLA-LES in Fig. 5. It is strongest during day-time. The bulk Richardson number criterion is also applied to the mean COSMO profiles and the resulting $z_i$ is shown as blue dashed line. The LES models produce a very similar boundary layer depth. Both models are lagging behind COSMO. As the LES are tied to the COSMO forcing, they also show peak heights close to COSMO. This behavior can partly

be attributed to the Newtonian relaxation which pulls the LES back towards the mean state given by the forcing.

Measurements from the three different major HOPE sites are taken into account for evaluating the performance of the LES models in terms of the boundary layer depth. The aerosol Raman lidar Polly$^{\mathrm{XT}}$ (Althausen et al., 2009, Polly hereafter) at LACROS site provides an estimate for the boundary layer depth based on the heights where the detected aerosols show a strong back-scatter signal. The Doppler wind lidar HALO (Schween et al., 2014) provides profiles of vertical velocity variance

at the JOYCE-site from which the boundary layer depth is deduced as the lowest height from the surface onwards where the vertical velocity variance is smaller than a threshold of $0.4\ \mathrm{m^2\ s^{-2}}$. As third data source 78 radio-soundings from KITcube site were used. The bulk Richardson number method was applied to the available soundings. In analyzing the soundings erroneous values near the surface were detected so the critical $\mathrm{Ri_b}$ is calculated from 100 m onwards. The applied criteria to the lidar data (vertical velocity variance and aerosol layer) are not boundary layer regime independent and usually work best for convective

boundary layer situations. Anyhow, they are a standard measurement product and the independent measurements of $z_i$ used in this study provide a general corridor for a representative boundary layer depth observed in the HOPE domain. Due to the different methods used to deduce the boundary layer depth, the aerosol lidar typically shows larger depths than the wind lidar (see Fig. 5) as the detected aerosol layers are a passive tracer for the boundary layer depth as compared to the dynamic criterion based on vertical velocity variance.

On most days, PALM, UCLA-LES and COSMO are able to reproduce the development of the boundary layer as the models lie inside the spread of the measurements resulting from surface heterogeneity and spatial variability of the boundary layer depth between the three sites. On days with strong vertical forcing, which are marked in light yellow colors in Fig. 5, the simulated peak depths agree less well with the observations. A day is characterized as a day with strong vertical forcing in case the prescribed larger-scale subsidence velocity averaged between 4 km and 8 km, denoted as $\widetilde{w_{\mathrm{SUB}}}$ in the following, is larger

than $50\ \mathrm{cm\ s^{-1}}$.





On 25 April, a day where shallow cumulus was observed but not simulated (see Tab. 2), the peak height is strongly under-estimated by the LES, but also by the host model COSMO. Overall, the daily development of the boundary layer depth can be qualitatively reproduced by both LES models.

### 4.1.3 Further boundary layer quantities

Figure 6 gives a further overview about the performance of the LES for boundary layer quantities like the surface sensible and latent heat fluxes, near surface wind direction, wind speed and potential temperature and the integrated water vapor IWV and liquid water path LWP. For the LES, the horizontal mean of the quantities is shown. At first glance, general agreement with observations is given. PALM and UCLA-LES are nearly indistinguishable apart from LWP. They are also rather close to COSMO

In the reference setup, potential temperature and humidity from the COSMO averaging box are prescribed homogeneously at the surface and the sensible and latent heat fluxes (shf and lhf) are calculated via Monin-Obukhov similiarity theory (see Sect. 2.3). These surface fluxes are very important as they directly determine the amount of energy input into the boundary layer. In Fig. 6a,b the surface fluxes from PALM and UCLA-LES are compared with the fluxes from the COSMO forcing and measurements from different energy balance stations. A total of five different stations located over different land-use classes in

close vicinity to the principal HOPE sites are taken into account (see Fig. 1b). From these measurements spatially representative values of the surface fluxes are derived and provided by Maurer et al. (2016). A weighted average (w.av.) of the five stations with the fraction of the respective land-use class in an area of 30 km × 30 km centered around KITcube site is calculated (see Maurer et al., 2016, for further details). In panels a and b, the weighted average is marked by purple stars. The fluxes at the individual stations show a considerable spread reflecting the large spatial variability for surface fluxes (heterogeneity) in the

HOPE region. By construction, shf and lhf in the LES are closely tied to the surfaces fluxes in the forcing and are also slightly lagging behind like the boundary layer depth. They agree roughly with the weighted average on most days. The peak shf in the LES and COSMO tends to be overestimated compared to the weighted average, whereas the lhf tends to be underestimated, especially for the last six days of the simulation period. Overall, the simulated surface fluxes can be seen as representative for the HOPE region.

For wind-engineering purposes, surface layer winds are very important. Measurements from the 120 m-meteorological tower at JOYCE site (Löhnert et al., 2015) and radio-soundings are compared to the LES and COSMO. The wind components $u$ and $v$ were linearly interpolated between the second and third prognostic level to obtain values for 120 m. All major changes in wind direction at a height of 120 m can be reproduced very well by the two LES models and COSMO (panel c). For the wind speed at 120 m height, the tower measurements and soundings show larger fluctuations than the models as the point measurements

contain turbulent signals which are smoothed out in the shown horizontal mean of the LES output. Taking these differences into account, the LES agree rather well with the wind speed observations.

The near-surface potential temperature at a height of 25 m from the JOYCE tower, the radio-soundings and the LES is depicted in panel e. For the LES the output at the first prognostic level is taken. PALM, UCLA-LES and COSMO are sys-





tematically too warm during the night. Overall, there is good agreement with observations although the amplitudes of the observations are slightly larger.

Observations of the column integrated quantities integrated water vapor IWV and liquid water path LWP, shown in panels f and g, are provided by the microwave radiometer HATPRO (Löhnert et al., 2015; Steinke et al., 2015) at JOYCE site. There

is good agreement for IWV between the LES, COSMO and HATPRO. Hence, the total amount of water vapor is accurately included into the LES by means of the larger-scale forcing method. The LWP (panel g) of the LES matches the observations better than COSMO despite the deficiency in terms of used warm microphysics. Anyhow, modeling LWP (which can be seen as proxy for clouds) with the long-term LES approach correctly is rather challenging.

The six days with strong vertical forcing (highlighted in yellow) show all rather high values of LWP in rough accordance with

HATPRO. As already discussed in section 4.1.1, there are 25 April and 1 May where shallow clouds could not be simulated although they have been observed which is also apparent in panel g. Furthermore, both LES differ more strongly compared to the previously discussed quantities as microphysics and numerics are closely tied and they are very important for allowing cloud formation in the LES.

## 4.2 Vertical structure

The main strength of LES is to resolve turbulence. To assess whether the long-term LES approach is able to produce realistic turbulence statistics, variance profiles for two distinct situations are discussed. The variances of vertical velocity $\overline{w'}^2$, potential temperature $\overline{\theta'}^2$ and specific humidity $\overline{q'_v}^2$ from PALM and UCLA-LES are compared to variance profiles from lidars located at KITcube site for a one hour period (11-12 UTC) for the clear-sky situation of 24 April and the shallow cumulus situation of 5 May. Three different lidars, namely the Doppler lidar WindTracer WTX combined with the Doppler lidar WLS7

(Maurer et al., 2016) from KIT, the rotational Raman lidar RRL (Behrendt et al., 2015) and the water vapor differential absorption lidar WVDIAL (Muppa et al., 2016) from University of Hohenheim were operated simultaneously during IOPs of HOPE, allows us to compare different lidar-based higher order moments with the LES to discuss the turbulence structure of the boundary layer on these two days.

Figure 7 shows the vertical velocity, potential temperature and specific humidity variances for 24 April and 5 May (11-

12 UTC). For 24 April, the lidar-based variances (solid purple lines) of the vertical velocity, the actual temperature and the absolute humidity were each recently published by Maurer et al. (2016); Behrendt et al. (2015) and Muppa et al. (2016). They also provide data for the cumulus-topped boundary layer of 5 May, which is analyzed for the first time in the present paper. The lidar turbulence signal at each height is calculated by subtracting the linear fit of the recorded time-series between 11 and 12 UTC from the original time-series. Based on this turbulence time-series, the variance for each record is calculated (see,

e.g., Behrendt et al., 2015). Note that the actual temperature variance as given by RRL was converted to potential temperature variance assuming a constant Exner function which was taken from the radio-sounding profile at 11 UTC of the respective day. The absolute humidity variance was converted similarly by means of the air density taken from the same sounding. In the cumulus case (5 May), the data points inside cloudy regions are not taken into account for the estimation of higher-order moments with RRL and WVDIAL. Furthermore, the potential temperature variance of RRL is only shown up to a height of



**Table 3.** Scaling values for 24 April 2013 and 5 May 2013 for 11-12 UTC (used in Fig. 7).

|  |  | PALM | UCLA-LES | Lidar (Kc) |
|---|---|---|---|---|
| 24 April 2013 | $z_i$ (m) | 1033 | 1091 | 1312 |
| 11-12 UTC | $\overline{w'\theta'_{vs}}$ (W m$^{-2}$) | 285.3 | 292.1 | 163.1 |
|  | $\overline{w'q'_{vs}}$ (W m$^{-2}$) | 168.3 | 156.0 | 129.6 |
|  | $w^*$ (m s$^{-1}$) | 2.022 | 2.075 | 1.810 |
|  | $\theta^*$ (K) | 0.117 | 0.117 | 0.075 |
|  | $q^*$ (g kg$^{-1}$) | 0.028 | 0.025 | 0.060 |
| 5 May 2013 | $z_i$ (m) | 1641 | 1465 | 1723 |
| 11-12 UTC | $\overline{w'\theta'_{vs}}$ (W m$^{-2}$) | 181.3 | 202.1 | 185.2 |
|  | $\overline{w'q'_{vs}}$ (W m$^{-2}$) | 160.9 | 140.6 | 127.5 |
|  | $w^*$ (m s$^{-1}$) | 2.037 | 2.031 | 2.053 |
|  | $\theta^*$ (K) | 0.076 | 0.084 | 0.077 |
|  | $q^*$ (g kg$^{-1}$) | 0.027 | 0.023 | 0.052 |

Values are averaged over one hour (11-12 UTC) on both days. Boundary layer depth $z_i$ in the
LES is determined based on the bulk-Richardson number criterion. For the lidar, $z_i$ is the top of
the aerosol layer based on backscatter signal. Surface buoyancy and latent heat fluxes, $\overline{w'\theta'_{vs}}$ and
$\overline{w'q'_{vs}}$ respectively, are horizontally averaged values in the LES and weighted averaged values
from the energy balance stations in case of lidar.

$0.7z_i$ which is near cloud base (see panel e) as the cloud layer is affected by saturation of the detector. In this case more noise
is found in the data and overlaps the true data thoroughly making the measurements less reliable.

Typically, higher-order moments from LES are deduced from a spatial (horizontal) average (e.g., Heinze et al., 2015) as
opposed to lidar measurements which define turbulence as departure from a temporal mean. To account for this difference,
variances from LES are shown in two different ways in Fig. 7. The solid black and green line denote the one-hour average of the
variances as defined by the departure from the horizontal mean (*hom*). Solid gray and light green areas show twice the standard
deviation resulting from the one-hour average of the slab-averaged variance profiles. Furthermore, virtual measurements were
conducted in the LES at four distinct locations which are equally spaced in the modeling domain. Grid-point data for four
independent columns (*colX*) with a high temporal resolution (30 s and 5 min for PALM and UCLA-LES, respectively) have
10 been saved. These time-series were used to calculate variances exactly as for the lidar data (detrending and temporal average
over one hour). These variance profiles are representative for a single-measurement inside the LES and are thus directly
comparable to the variances deduced from lidar. They are depicted as thin dashed black and green lines in Fig. 7.

To account for a better comparison between observed and simulated variances, all profiles in Fig. 7 are scaled (non-
dimensionalized) by means of the free convective Deardorff (1970a) scales. These are the convective velocity scale $w^* = \left( \frac{g}{\theta_{vs}} \overline{w'\theta'_{vs}} z_i \right)^{\frac{1}{3}}$,
the convective temperature scale $\theta^* = \frac{\overline{w'\theta'_{vs}}}{w^*}$ and the convective humidity scale $q^* = \frac{\overline{w'q'_{vs}}}{w^*}$, where $\overline{w'\theta'_{vs}}$ denotes the kinematic
surface buoyancy flux and $\overline{w'q'_{vs}}$ is the kinematic surface latent heat flux (see Tab. 3). The vertical axis (height) is normalized



**Table 4.** Simulated and observed cloud boundaries on 5 May 2013 for 11-12 UTC.

|  | PALM | UCLA-LES | Cloudnet (LA) | Cloudnet (JO) | ceilometer (JO) |
|---|---|---|---|---|---|
| $z_{cb}$ (m) | $1333 \pm 40$ | $1294 \pm 43$ | $1464 \pm 112$ | $1546 \pm 178$ | $1365 \pm 49$ |
| $z_{ct}$ (m) | $1721 \pm 83$ | $1713 \pm 62$ | $1594 \pm 118$ | $1735 \pm 196$ | $1526 \pm 56$ |
| $d_c$ (m) | $388 \pm 58$ | $419 \pm 33$ | $133 \pm 60$ | $189 \pm 146$ | $171 \pm 53$ |
| $N$ | 13 | 13 | 33 | 21 | 125 |

Values include mean and standard deviation over 11-12 UTC. Cloud boundaries in LES are determined based on
horizontally averaged profiles of cloud liquid water. Cloud base height, cloud top height and cloud layer depth are
denoted by $z_{cb}$, $z_{ct}$ and $d_c$ respectively. The number of samples entering the averaging period is $N$. See Tab. 1 for an
overview of the observations used.

by means of the boundary layer depth. For all lidar-derived profiles, the boundary layer depth is determined by estimating the
top of the aerosol layer from lidar backscatter data (method 2 in Maurer et al., 2016). The required surface fluxes are taken
from the weighted average of five different energy balance stations (see also Sect. 4.1) which is, based on Maurer et al. (2016),
representative for a larger area. The LES-based scaling values are derived from the bulk-Richardson number-based $z_i$ and the
one-hour average of the horizontal mean surface buoyancy and latent heat flux. All values are summarized in Table 3.

     Comparing the horizontal mean variances of PALM and UCLA-LES in Fig. 7 generally, we note that they both show a
very similar vertical structure. In all six cases variances from PALM are slightly larger than variances from UCLA-LES which
becomes most prominent for the peak values of the scalar variances at the top of the boundary layer (panels b, c, e and f). The
differences in variances between PALM and UCLA-LES are in the same order as discussed in several LES intercomparison
studies (e.g. Stevens et al., 2001; Siebesma et al., 2003; Stevens et al., 2005). It can be attributed to different numerics like the
advection scheme. As UCLA-LES uses a monontone scheme for the scalars and PALM not (see also Sect. 2.1), fluctuations
are damped more strongly resulting in slightly less variances (turbulence).

     On 24 April around noon, the boundary layer is cloud-free, well mixed and topped by a capping inversion as seen by radio-
sounding profiles in Fig. 3c. The LES reproduce this structure which manifests also in the variance profiles (Fig. 7a-c). The
LES-based vertical velocity variances reveal the typical peak around $0.3z_i$ and decrease monotonically above. The vertical
velocity variance from Doppler lidar exhibits a maximum at around $0.5z_i$ and shows a second smaller peak around $0.9z_i$. A
longer averaging period of about 3 h would lead to a decrease of the height of the lower maximum to about $0.3z_i$ (Maurer et al.,
2016) which emphasizes that the chosen averaging time might be too small to receive robust $\overline{w'^2}$ statistics comparable to LES.
This is confirmed by the large differences between the given virtual measurements. The horizontal mean profiles and to a larger
extent also the virtual measurements are inside the uncertainty range of the Doppler lidar. Nonetheless, it should be kept in
mind that a departure of the horizontally averaged LES variances from the lidar variances does not necessarily mean that the
LES variances are not representative as the statistical error based on Lenschow et al. (1994) does not always show how large
the uncertainties really are - especially in case of heterogeneous surfaces (Sühring and Raasch, 2013).





The LES-based scalar variances show their distinct maxima on 24 April at the top of the boundary layer (Fig. 7b,c) where warmer and less humid tropospheric air is entrained producing large turbulent fluctuations. This is principally in accordance with the lidar measurements. The peak values of the lidar-based scalar variances are significantly higher than the ones of the LES - even when taking the virtual measurements in the LES models into account. Here, it becomes apparent that the vertical

grid-spacing of 50 m used in LES is much too coarse to sufficiently resolve the strong vertical gradients at the boundary layer top. Another reason for the underestimation of scalar variance peak values might also be the usage of homogeneous surface forcing which allows only to prescribe surface forcing representative for the larger area which might not necessarily be similar to the forcing actually present at the measurement site. The specific humidity variance from WVDIAL shows a rather unusual lower peak at around $0.85z_i$ (panel c) which Muppa et al. (2016) associate with entrainment of an elevated humidity layer into

the convective boundary layer. The second peak in vertical velocity variance at around $0.9z_i$ might be also associated with this event.

On 5 May a shallow cumulus layer was observed at JOYCE site and simulated around noon (see Fig. 3g,h). The mean profiles of potential temperature and specific humidity of PALM and the radiosoundings barely show the existence of the cloud layer as it is rather shallow. Table 4 provides an overview of the observed and simulated cloud boundaries between 11 and

12 UTC. An average of Cloudnet observations at LACROS and JOYCE and a ceilometer at JOYCE site results in a 156 m deep layer. The cloud layer in both simulations is about 2.5 times deeper with about 388 m for PALM and 419 m for UCLA-LES. The LES are expected to show deeper cloud layers as the maximum height of a sampled cloud in the domain determines the depth whereas the measurements sample at one point only. Both LES simulate a total cloud cover during noon that is not higher than 5 % (not shown) and also the LWP does not show a significant signal (see Fig. 6g) supporting the finding of a very weak

shallow cumulus layer in the models. The cloud boundaries are also depicted in Fig. 7d,e,f as gray and green dashed layer for the LES. The cloud boundaries from observations at KITcube are not shown as it was not possible to reliably estimate them from the lidars at KITcube site. There have been only 4 tiny clouds passing the lidars during the one-hour period (not shown). Note that the cloud layers are also scaled which might lead to a different impression while comparing the thicknesses.

The variances on 5 May also show no distinct feature of a well developed cumulus layer on top of a well-mixed sub-

cloud layer in the LES as well as in the observations. Their shapes resemble strongly those of the variances in the cloud-free convective boundary layer discussed before. For the vertical velocity variance, the LES horizontal mean as well as most of the virtual measurements are close to the uncertainty range of the lidar showing also a similar shape as the lidar. The potential temperature variance can only be compared below $0.7z_i$ as it is not available from RRL higher above. LES and lidar both show low variances in the well-mixed part of the boundary layer. The maximum of specific humidity variance is located slightly

higher than those of the LES.

Overall, the long-term LES approach is able to deliver variance (turbulence) profiles that are in a satisfactory agreement with lidar observations.





**Table 5.** Parameters of the simulated cases.

| Case | LES model | $\Delta$ | $L_1 \times L_2$ | $N_1 \times N_2 \times N_3$ | $z_0$ | $t_{sim}$ | $\tau$ | $L_{COSMO}$ | $\mathcal{T}_{COSMO}$ | surface BC |
|------|-----------|----------|------------------|------------------------------|-------|-----------|--------|-------------|----------------------|------------|
|      |           | (m)      | (km)             |                              | (m)   | (d)       | (h)    | (°)         | (h)                  |            |
| RP | PALM | 50 | $48 \times 48$ | $960 \times 960 \times 144$ | 0.45 | 19 | 6 | 2.0 | 3 | prescr. $\theta$ and $q_v$ |
| RU | UCLA | 50 | $48 \times 48$ | $960 \times 960 \times 144$ | 0.45 | 19 | 6 | 2.0 | 3 | prescr. $\theta$ and $q_v$ |
| RPS | PALM | 50 | $4.8 \times 4.8$ | $96 \times 96 \times 144$ | 0.45 | 19 | 6 | 2.0 | 3 | prescr. $\theta$ and $q_v$ |
| RUS | UCLA | 50 | $4.8 \times 4.8$ | $96 \times 96 \times 144$ | 0.45 | 19 | 6 | 2.0 | 3 | prescr. $\theta$ and $q_v$ |
| F0.25 | PALM | 50 | $4.8 \times 4.8$ | $96 \times 96 \times 144$ | 0.27 | 3 | 6 | 0.25 | 3 | prescr. $\theta$ and $q_v$ |
| F0.5 | PALM | 50 | $4.8 \times 4.8$ | $96 \times 96 \times 144$ | 0.31 | 3 | 6 | 0.5 | 3 | prescr. $\theta$ and $q_v$ |
| F1.0 | PALM | 50 | $4.8 \times 4.8$ | $96 \times 96 \times 144$ | 0.41 | 3 | 6 | 1.0 | 3 | prescr. $\theta$ and $q_v$ |
| F3.0 | PALM | 50 | $4.8 \times 4.8$ | $96 \times 96 \times 144$ | 0.44 | 3 | 6 | 3.0 | 3 | prescr. $\theta$ and $q_v$ |
| F4.0 | PALM | 50 | $4.8 \times 4.8$ | $96 \times 96 \times 144$ | 0.40 | 3 | 6 | 4.0 | 3 | prescr. $\theta$ and $q_v$ |
| TR1 | PALM | 50 | $4.8 \times 4.8$ | $96 \times 96 \times 144$ | 0.45 | 3 | 6 | 2.0 | 1 | prescr. $\theta$ and $q_v$ |
| Nno | PALM | 50 | $4.8 \times 4.8$ | $96 \times 96 \times 144$ | 0.45 | 19 | $\infty$ | 2.0 | 3 | prescr. $\theta$ and $q_v$ |
| N1 | PALM | 50 | $4.8 \times 4.8$ | $96 \times 96 \times 144$ | 0.45 | 19 | 1 | 2.0 | 3 | prescr. $\theta$ and $q_v$ |
| N12 | PALM | 50 | $4.8 \times 4.8$ | $96 \times 96 \times 144$ | 0.45 | 19 | 12 | 2.0 | 3 | prescr. $\theta$ and $q_v$ |
| FLX | PALM | 50 | $4.8 \times 4.8$ | $96 \times 96 \times 144$ | 0.45 | 19 | 6 | 2.0 | 3 | prescr. fluxes |
| RPS12.5 | PALM | 12.5 | $4.8 \times 4.8$ | $384 \times 384 \times 480$ | 0.45 | 3 | 6 | 2.0 | 3 | prescr. $\theta$ and $q_v$ |
| RPS25 | PALM | 25 | $4.8 \times 4.8$ | $192 \times 192 \times 266$ | 0.45 | 3 | 6 | 2.0 | 3 | prescr. $\theta$ and $q_v$ |
| RPS100 | PALM | 100 | $4.8 \times 4.8$ | $48 \times 48 \times 84$ | 0.45 | 3 | 6 | 2.0 | 3 | prescr. $\theta$ and $q_v$ |

$\Delta$ denotes the grid spacing, $L_1$, $L_2$ are the model domain sizes in $x_1$ and $x_2$ directions, respectively, $N_1$, $N_2$ and $N_3$ are the number of grid points in $x_1$, $x_2$ and $x_3$ directions, respectively, $z_0$ is roughness length for momentum, $t_{sim}$ is the simulation time, $\tau$ is the relaxation time-scale, $L_{COSMO}$ is the averaging domain size of the larger-scale forcing data (given in degrees on the geographical grid), $\mathcal{T}_{COSMO}$ is the temporal resolution of the larger-scale forcing data. The abbreviations *surface BC* and *prescr.* stand for for surface boundary conditions and prescribed, respectively.

## 5 Sensitivities

To study how robust the previously discussed results are with respect to the chosen setup, the reference simulations *RP* and *RU* were complemented by 14 additional simulations with PALM. Table 5 lists the simulations with their differences in the setups relative to the setup *RP/RU*, which was described in Sect. 2.3. Most of these additional simulations were run on a smaller horizontal domain ($4.8 \times 4.8$ km$^2$ instead of $48 \times 48$ km$^2$) and for the first three days only (24-26 April) for the sake of computational resources (Note that *RP* and *RU* ran on 2000 cores for around 7 and 10 days, respectively). The period 24-26 April was chosen as it contains three different boundary layer states (clear-sky, shallow clouds, frontal passage) in a row, being a condensed representative of the longer period.




To compare all the experiments a metric based on the boundary layer depth (see Fig. 5) is constructed. As $z_i$ is a central quantity for evaluating mean boundary layer characteristics, it is chosen as basis for the metric. For each available value, the absolute difference in boundary layer depth of PALM between the host model COSMO, the aerosol lidar Polly and the wind lidar HALO, respectively, are calculated. Then, an average over the number of available daily time-spans from 12-14 UTC

(either 19 or 3 depending on the case) is taken and the standard deviation is provided accordingly. This metric is called *mean peak difference to PALM* in the following. A daily averaging time-span of two hours (12-14 UTC) was chosen to consider the state of a well developed boundary layer in a quasi-steady period. Figure 8 shows the mean peak difference in boundary layer depth to PALM for all the additional simulations. At a first glance it can be noted that the mean peak differences to PALM of COSMO, Polly and HALO in most cases show the same behavior. The metric based on wind lidar HALO usually shows the

highest and positive values meaning that the peak boundary layer depth of PALM is usually higher than the one measured by HALO.

Comparing the 19-days reference simulation *RP* with the 19-days simulation RPS, which was conducted on the small horizontal domain, we note that the domain size has virtually no effect on the mean peak difference to PALM (Fig. 8a, comparing cases *RP* and *RPS (19d)*). Thus, robust first-order statistics are gained even in case the domain size is significantly smaller than

in the reference case. This suggests that the internally generated (non-forced) mesoscale circulations, which can develop on a $50\,\mathrm{km} \times 50\,\mathrm{km}$ domain, are not particularly important.

A fundamental parameter of the larger-scale forcing method is the averaging domain size for the applied forcing data $L_{\mathrm{COSMO}}$, specified in degrees on the geographical grid (see also Appendix A). For the reference runs *RP* and *RU*, a size of $L_{\mathrm{COSMO}} = 2.0°$ was used. To evaluate whether the size of the averaging box is appropriate to represent larger-scale processes,

the simulations *F0.25*, *F0.5*, *F1.0*, *F3.0* and *F4.0* (see Tab. 5) were conducted, where the COSMO averaging domain sizes were varying from $0.25°$ to $4.0°$ which corresponds to horizontal extensions $\mathcal{D}_x \times \mathcal{D}_y$ of $17.5 \times 27.8\,\mathrm{km}^2$ to $280 \times 444\,\mathrm{km}^2$ being equivalent to averaging over $10 \times 10$ to $160 \times 160$ COSMO grid points. The averaging domain size of the COSMO forcing has a large impact on the boundary layer depth as can be seen in Fig. 8b. Especially the two smallest averaging domain sizes produce large discrepancies in peak boundary layer depth to the estimates of $z_i$ stemming from Polly and HALO lidar. Thus, as the av-

eraging area gets small, more mesoscale flows, which COSMO does not necessarily well represent, are sampled. Nonetheless, mean boundary layer characteristics become less sensitive if a $2.0°$ averaging domain size or larger is used. Cloud structures and precipitation depend more strongly on the averaging domain size of the forcing (not shown). Overall, the averaging domain should have a size which is large enough to not include mesoscale fluctutations on the one side and which is small enough to still account for a localized, representative area like the HOPE region.

The temporal resolution of the forcing data is 3 h which also includes the prescribed surface temperature and humidity and via Monin-Obukhov similiarity theory the surface fluxes. However, boundary layer time-scales are usually much shorter (the turnover time-scale is about 10 minutes around noon for the presented period). As the simulations are strongly determined by the imposed surface fluxes, the question arose whether prescribing new surface values every 3 h is too infrequent to impose the signal of a proper diurnal cycle. Thus, the simulation *TR1* was performed, where forcing data with a temporal resolution of

1 h was used. As the larger-scale horizontal and vertical advective forcing act on larger time-scales than the surface forcing, a





higher temporal resolution should affect the surface fluxes most. Comparing the cases *RPS (3d)* and *TR1* shown in Fig. 8a, it can be noticed that the metrics are nearly identical. The higher temporal resolution seems to bring no additional value. Hence, it is concluded that a 3-hourly forcing data set is sufficient to impose a proper diurnal cycle in the simulations.

As nudging (Newtonian relaxation) does not represent a real physical process (Randall and Cripe, 1999), it was analyzed how

crucial the results depend on the nudging time-scale and on the nudging itself. Three additional simulations were performed where a stronger nudging with $\tau = 1$ h (case *N1*), a weaker nudging with $\tau = 12$ h (case *N12*) and no nudging at all ($\tau \to \infty$, case *Nno*) compared to the reference nudging time-scale of 6 h were used. The simulation without nudging can also be interpreted as a simulation were the radiative forcing is completely switched off as the effect of radiation is indirectly mimicked via the relaxation (see Sect. 2.3). The mean peak difference to PALM (Fig. 8c) shows only a weak dependence for $\tau \leq 12$ h. In

case Newtonian relaxation is completely turned off, the mean peak difference to PALM increases strongly. In this case PALM strongly overestimates the boundary layer depth compared to the forcing and the observations. The overall performance of the simulation becomes worse. This analysis shows that using nudging with reasonable nudging time-scales of several hours is beneficial for the long-term LES framework. Furthermore, the mean boundary layer characteristics barely depend on the actual choice of the nudging time-scale supporting the robustness of the setup.

To test the impact of the individual larger-scale forcing components, several tests were made in which the forcing components were mutually switched off and then added one after the other (not shown). These tests suggested that all components should be used in combination for obtaining the best results with respect to the observations. This is in agreement with the single-column model study of Sterk et al. (2015) where they studied the realistic simulation of clear-sky stable boundary layers over snow-covered surfaces.

In the reference setup, Dirichlet conditions are used at the surface meaning that potential temperature and specific humidity are prescribed at the surface. The alternative is to prescribe surface fluxes directly (using Neumann boundary conditions). The latter was used in case *FLX*. Overall, the prescribed surface fluxes are slightly smaller and show a time lag in respect to the fluxes which are calculated in case *RPS (19d)* (not shown). Comparing the cases *RPS (19d)* and *FLX* concerning the mean peak difference to PALM (Fig. 8a) it can be seen that the metric for COSMO changes only marginally and that the metric for

Polly deteriorates whereas the metric for HALO improves. Taking also the arguments of Basu et al. (2008) into account that for modeling stable boundary layers prescribing surface fluxes should be avoided, we think prescribing surface values is the better option, as during the conducted multiple-day LES stable regimes are simulated to a considerable fraction.

To evaluate the influence of the numerical grid spacing, the three-day simulation *RPS (3d)* with an isotropic grid spacing $\Delta = 50$ m was rerun using two finer grid spacings ($\Delta = 25$ m called *RPS25* and $\Delta = 12.5$ m called *RPS12.5*) and one coarser

grid spacing $\Delta = 100$ m called *RPS100*). Only minor differences were observed between the runs in the time-series of the boundary layer depth which mainly occur during nighttime. This indicates, that the differences between the runs are closely linked to their different capabilities of resolving the shallow stable boundary layer at night. The influence on the better resolved nighttime stable boundary layer on the following convective day is rather small as van Stratum and Stevens (2015) already showed. The simulated clouds also do not show any dependence on the grid-spacing. Figure 8d shows in terms of the mean

peak difference metric that the influence of the grid spacing on mean boundary layer characteristics is negligible.





## 6   Summary and conclusions

In this study long-term LES with PALM and UCLA-LES are evaluated to answer the question if LES in a semi-idealized setup are able to simulate boundary layer characteristics and turbulence in a realistic manner. The semi-idealized approach consists of using periodic lateral boundary conditions and a homogeneous surface forcing together with prescribing time-dependent

larger-scale forcing and nudging deduced from the mesoscale numerical weather prediction model COSMO to account for the synoptic conditions at a specific location. A continuous period of 19 days of the HOPE measurement campaign is chosen and the simulation results are compared to the multi-sensor HOPE data set. The three principal measurement sites of HOPE allow to obtain a more representative view on the larger observational area. This circumstance facilitates the comparison to the LES which, by construction, can only deliver a flow which is representative for the HOPE region. The analysis focuses on key

boundary layer quantities like the boundary layer depth, near surface temperatures and winds, integrated quantities like IWV and LWP and turbulence statistics in terms of variance profiles. A metric based on the peak boundary layer depth is used to compare several sensitivity runs. With these additional simulations the robustness of the reference setup is investigated.

The (unphysical) nudging tendency, which prevents model drift in time, is less important compared to larger-scale horizontal and vertical advective tendencies. In cases with strong larger-scale forcing, also the nudging tendencies are significant.

The reference simulation shows reasonable agreement with the HOPE measurements. The principal character of the day (weather situation) can be reproduced by the LES in about 80 % of the cases. Simulating cloud-topped boundary layers correctly is a challenge for the long-term LES. The daily development of the boundary layer depth is in principal agreement with lidar measurements. The LES surfaces fluxes are in a rough agreement with the weighted averaged surface fluxes in the HOPE area showing that the surface forcing is representative for the HOPE area. Both LES models used produce very similar

results.

The LES models seem to track COSMO closely and deviate from the observations in a similar fashion as COSMO does. This can be interpreted in two ways. Either deviations from the observations are inherited from the host model or they represent the signature of mesoscale forcing that the present approach is incapable of capturing. By using LES in a more realistic setup with open boundary conditions these hypotheses might be tested.

LES turbulence statistics in terms of variance profiles are in a satisfactory agreement with lidar measurements during HOPE. The peak in scalar variances at the top of the boundary layer is underestimated by LES indicating that presumably the resolution used in the LES is rather coarse for correctly representing strong gradients, and that heterogeneity is missing.

The chosen semi-idealized setup is rather robust and insensitive to the horizontal domain size, the grid spacing, the temporal resolution of the forcing data and the surface boundary condition in terms of mean boundary layer characteristics. Thus, the

internally generated mesoscale circulation on a larger domain are not particularly important and the character of the biases is not strongly dependent on the model or how the forcing is applied. There is a dependence on the averaging size of the forcing data. If the averaging domain is large enough and mesoscale fluctuations are sufficiently filtered out, the results converge. Using nudging itself to prevent model drift in time is important. The actual value for the relaxation time-scale is of minor importance provided that it is in the order of several hours.





As the semi-idealized setup stably represents a wide range of observed weather situations, it is also applicable as superparameterization (Grabowski, 2016) in a global model. It would be interesting to study how the overall performance of a global model with superparameterization depends on the chosen grid size which is tied to the horizontal domain size of the imbedded LES. As the LES obtain mean forcing profiles from the global model, the overall domain size from which the forcing is constructed might play a role as the semi-idealized setup depends on the averaging size of the forcing data.

The long-term LES approach cannot only be used to simulate periods at meteorological super-sites like in Schalkwijk et al. (2015) but also for simulating periods of (or even whole) measurement campaigns to support the interpretation of measurement results. This approach is for example also followed in the Large-Eddy Simulation (LES) ARM Symbiotic Simulation and Observation (LASSO) project (http://www.arm.gov/science/themes/lasso) where continuous LES of the Southern Great Plains atmospheric radiation measurement (ARM) supersite are under development.

One strength of the semi-idealized approach is that it is able to deliver robust turbulence statistics and a good representation of clouds, as typical for LES, and that it accounts for a localized area responding to every-day weather. However, a certain variability coming from the heterogeneous surface which usually surrounds any real observational site is neglected in the LES. The semi-idealized long-term LES approach can also be seen as an intermediate step towards LES in an limited-area setup, where, for example, a land-surface model and interactive radiation are used. In the framework of HD(CP)$^2$, these kind of simulations are performed over Germany. They are compared to the semi-idealized simulations presented here and the HOPE data set in Heinze et al. (2016).

## Appendix A:  Construction of forcing data

To filter out any impact of small scale phenomena in the forcing data, the used COSMO (Baldauf et al., 2011) analysis data (with a spatial and temporal resolution of 2.8 km and 3 h, respectively) is averaged spatially. Note that the semi-idealized LES approach requires vertical profiles of geostrophic wind components $u_{g,i}$, of larger-scale velocity vector $u_{i,LS}$, of liquid water potential temperature $\theta_{l,LS}$, of total water specific humidity $q_{t,LS}$ and of larger-scale gradients (horizontal and vertical) of $\theta_{l,LS}$ and $q_{t,LS}$ (see Eqn. (1)-(4)). Moreover, corresponding surface conditions of temperature, humidity (or the respective sensible and latent heat fluxes) and hydrostatic pressure (which is important for cloud microphysics) are needed.

First, a spatial averaging domain with side lengths $\mathcal{D}_x$ (zonal) and $\mathcal{D}_y$ (meridional) is defined. These side lengths should be large enough to filter the small scales (see Sect. 5 for a discussion of adequate averaging domain sizes). For determining the entire set of larger-scale quantities required for the long-term LES approach, five averaging domains are needed, as shown in Fig. 9:

- one centered domain (black square) for the determination of surface conditions and vertical profiles of $u_{g,i}$, $u_{i,LS}$, $\theta_{l,LS}$ and $q_{t,LS}$ and

- four shifted domains (red and blue squares) for the determination of larger-scale horizontal gradients of $\theta_{l,LS}$ and $q_{t,LS}$.




The averaged quantities of the centered domain are then assumed to represent the large scale quantities in the LES. The centers of the shifted domains are located one-half $\mathcal{D}_x$ in the east-west direction and one-half $\mathcal{D}_y$ in the north-south direction, respectively. Hence, the larger-scale gradients used in Eq. (2) are approximated as follows:

$$\frac{\partial \varphi_{\mathrm{LS}}}{\partial x_1} = \frac{\varphi_{\mathrm{LS,east}} - \varphi_{\mathrm{LS,west}}}{\mathcal{D}_{\mathrm{x}}} \tag{A1}$$

$$\frac{\partial \varphi_{\mathrm{LS}}}{\partial x_2} = \frac{\varphi_{\mathrm{LS,north}} - \varphi_{\mathrm{LS,south}}}{\mathcal{D}_{\mathrm{y}}}, \tag{A2}$$

since the averaged quantities are assumed to represent the larger-scale conditions at the center of each domain.

### Data availability

Primary data and scripts used in the analysis and other supplementary information that may be useful in reproducing the author's work are archived by the Max Planck Institute for Meteorology and can be obtained at http://hdl.handle.net/.... (Heinze, 2016).

*Acknowledgements.* This study was supported by the Federal Ministry of Education and Research in Germany (Bundesministerium für Bildung und Forschung; BMBF) through the research program *High Definition Clouds and Precipitation for Climate Prediction - HD(CP)$^2$*. The simulations were performed on the Cray XC30/40 of *The North-German Supercomputing Alliance* (HLRN), Hannover and Berlin, Germany and on the IBM Power6 of *The German Climate Computing Center* (DKRZ), Hamburg, Germany. The NCAR command language (version 6.3.0, http://dx.doi.org/10.5065/D6WD3XH5) was used for analysis and visualization.

We thank D. Klocke for providing the COSMO larger-scale forcing data, R. Neggers for a discussion about how to derive the larger-scale forcing data from COSMO, H. Knoop for his support in generating the volume rendered visualization used in Fig. 3 with VAPOR (www.vapor.ucar.edu), M. Schmidt for providing the surfaces fluxes from the TERENO sites, H. Baars for providing the boundary layer depth data from aerosol lidar Polly$^{\mathrm{XT}}$, U. Löhnert for providing the boundary layer depth data from the wind lidar HALO and Cloudnet data for JOYCE, P. Seifert for providing the Cloudnet data for JOYCE, A. Knaps for providing the data from the meteorological tower at JOYCE, A. Lammert-Stockschläder and V. Grützun for continuous support with the measurement data-sets through the HD(CP)$^2$ observational data portal (https://icdc.zmaw.de/hdcp2.html), the HD(CP)$^2$ teams of KIT Karlsruhe and University of Cologne for launching radiosondes and Alberto de Lozar for comments on an earlier version of the manuscript.



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





**Figure 3.** Snapshots and mean profiles of four different days (24 April, 26 April, 5 May, 10 May) taken at 11 UTC. The left column (panels a,d,g,j) shows images taken with the total sky imager TSI-880 at JOYCE site, the middle column (panels b, e, h, k) shows volume-rendered cloud water specific humidity $q_c$ of the PALM reference simulation $RP$ and the right column (panels c, f, i, l) shows mean profiles of specific humidity $q_v$ (black)), potential temperature $\theta$ (red), cloud and rain water specific humidity $q_c$ (blue) and $q_r$ (light blue), respectively. The solid lines are horizontally averaged profiles of $RP$ and the dashed lines are profiles from radio soundings (radios.) launched at KITcube site. Note that the vertical axis in panel f extends up to 10 km.





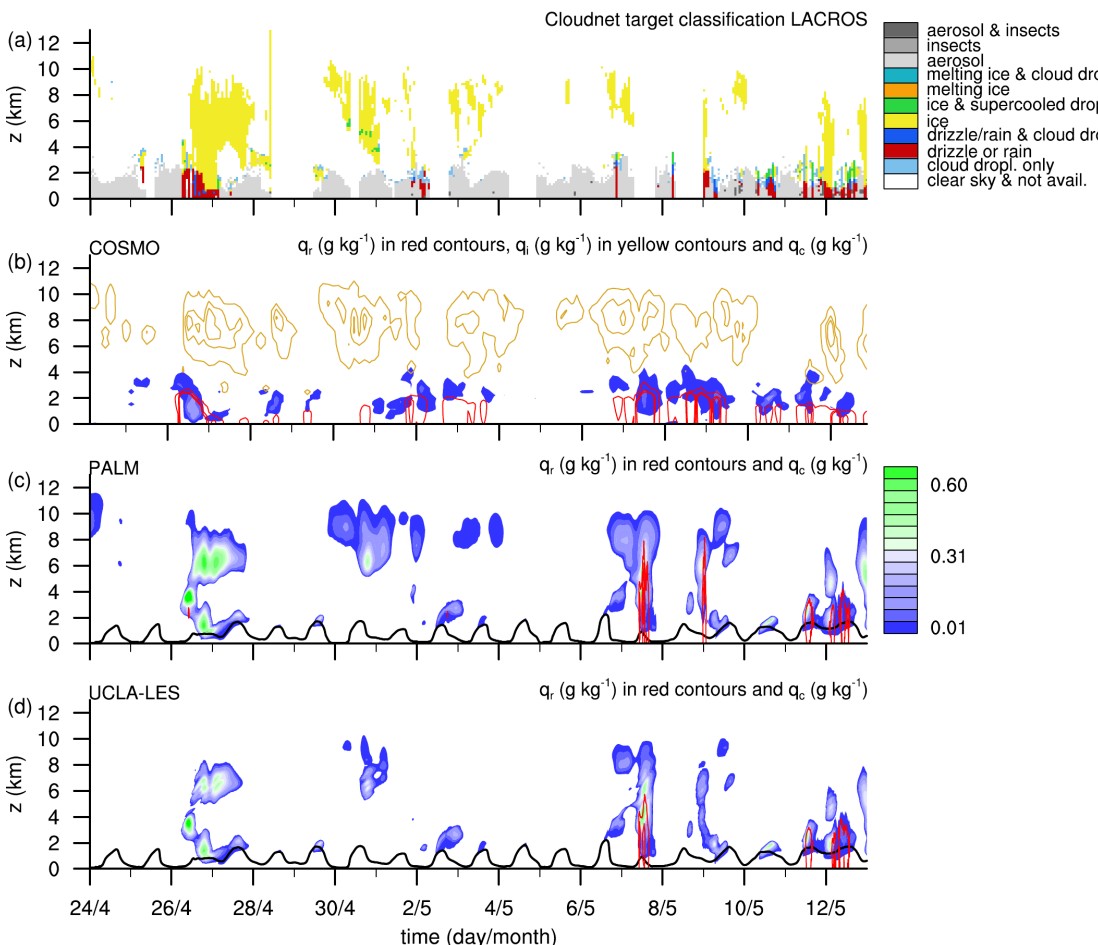

**Figure 4.** Time-height cross-sections of Cloudnet target classification (a), specific cloud ice $q_i$ (in yellow contours ranging from 0.001 – 0.21 g kg$^{-1}$ by 0.1 g kg$^{-1}$), specific rain water $q_r$ (in red contours ranging from 0.001 – 0.01 g kg$^{-1}$ by 0.05 g kg$^{-1}$) and specific cloud water $q_c$ (colored contours) of the COSMO forcing in panel b, specific cloud and rain water of PALM (run *RP*) in panel c and specific cloud and rain water of UCLA-LES (run *RU*) in panel d. The red contours in panels c and d have the same values as in panel b. The black lines in panels c and d denote the boundary layer depth according to the bulk-Richardson number criterion (see also Fig. 5). The same colorbar is used in panels b, c and d. Note, that in panels b, c and d the time series of horizontally averaged profiles are shown.





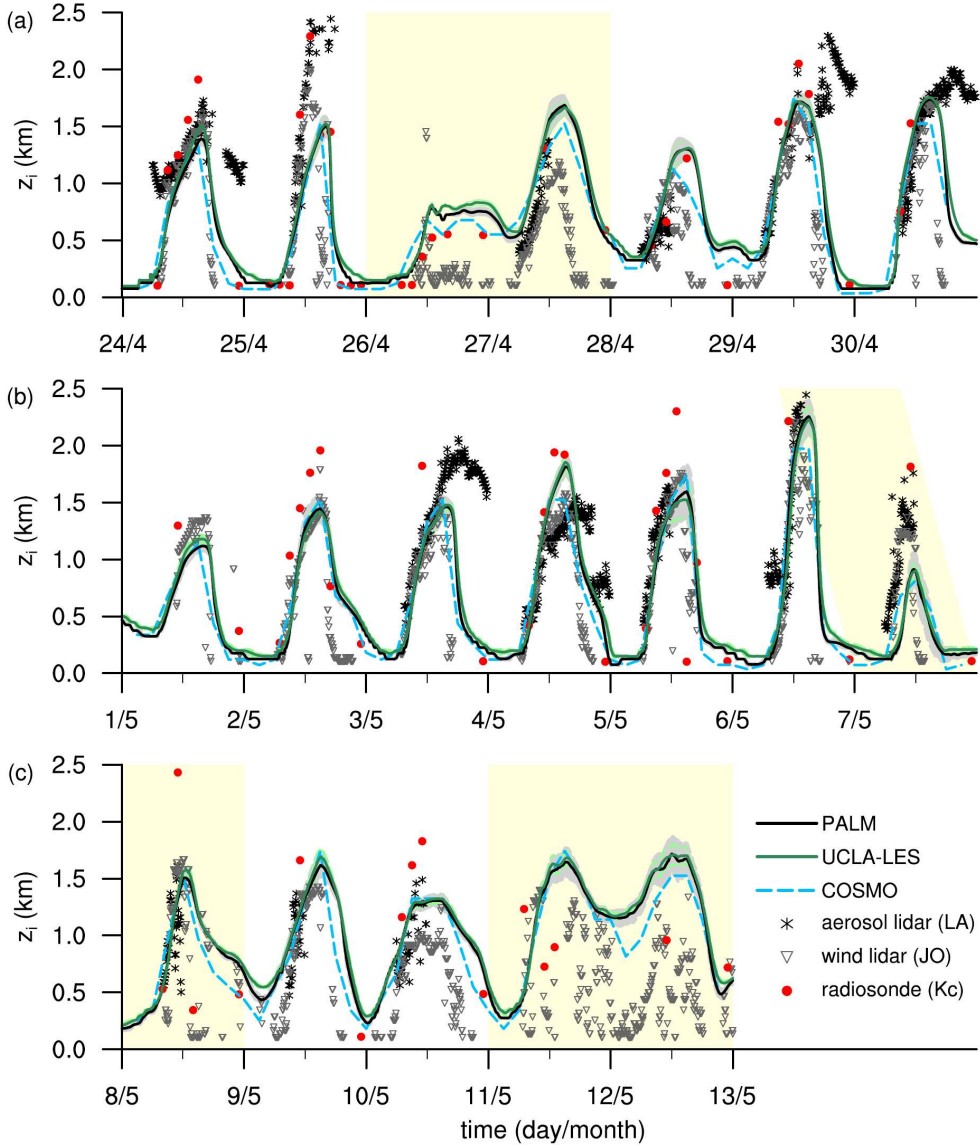

**Figure 5.** Temporal evolution of the boundary layer depth $z_i$ for the total 19-day period (grouped in weeks). $z_i$ is determined by means of the bulk-Richardson number criterion in all three models (PALM, UCLA-LES and COSMO) and in the radiosonde data. A criterion based on the vertical velocity variance and detected aerosol layers is used for the wind lidar and aerosol lidar, respectively. Radiosondes were launched at the KIT-cube site, the wind lidar and aerosol lidar took measurements at JOYCE and LACROS site, respectively. Gray and green shading denote twice the standard deviation of $z_i$ in PALM and UCLA-LES, respectively. Yellow highlighting marks days with strong vertical forcing ($\widetilde{w_{\mathrm{SUB}}} > 0.05$ m s$^{-1}$).





**Figure 6.** Temporal evolution of surface sensible heat flux (shf panel a), surface latent heat flux (lhf panel b), wind direction at 120 m height (wdir$_{120m}$ panel c), wind speed at 120 m height ($|v|_{h,120m}$ panel d), potential temperature at 25 m height ($\theta_{25m}$ panel e), integrated water vapor (IWV panel f), and liquid water path (LWP panel g). An overview of the measurements and abbreviations is given in Tab. 1. The top legend refers to panels a and b, the middle panel refers to panels c, d and e and the lower legend refers to panels f and g. Gray and green shading in panel g denote twice the standard deviation of LWP in PALM and UCLA-LES, respectively. Yellow highlighting marks days with strong vertical forcing ($\widetilde{w_{SUB}} > 0.05$ m s$^{-1}$).



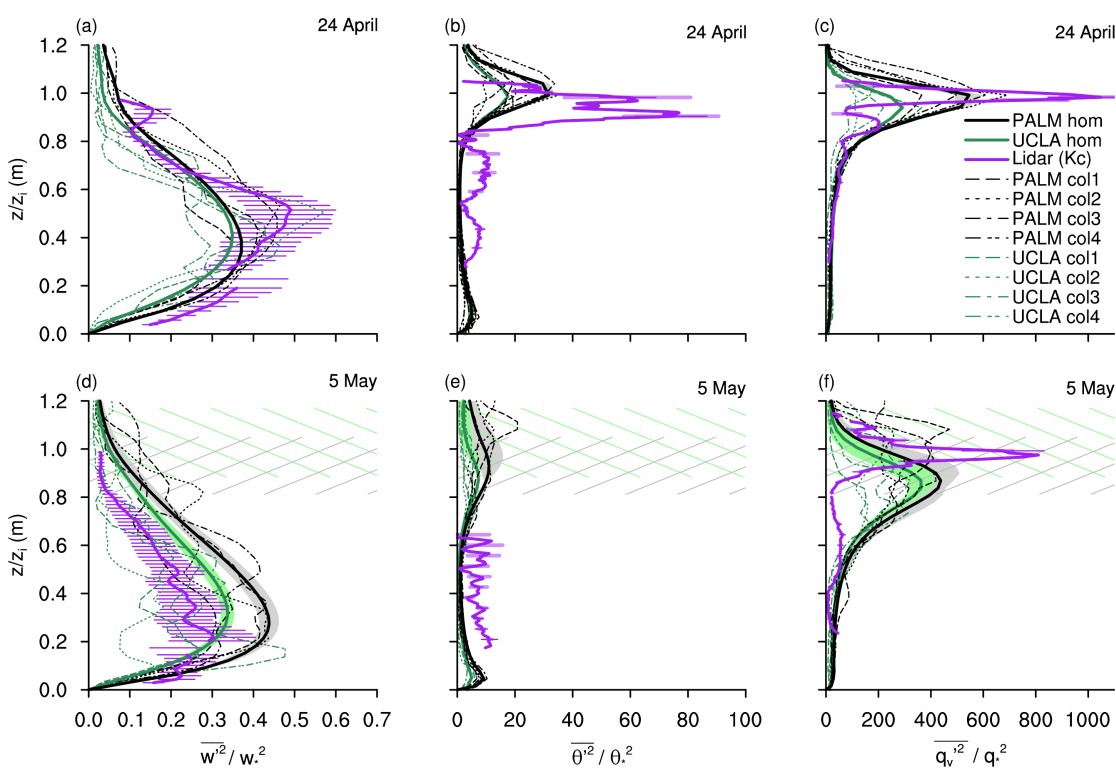

**Figure 7.** Normalized vertical profiles of vertical velocity variance, panels a and d, potential temperature variance, panels b and e and specific humidity variance, panels c and f, for a 1 h-period between 11 and 12 UTC for 24 April and 5 May 2013, respectively. Solid black and green lines show variances of PALM and UCLA-LES determined as departure from the horizontal mean (*hom*) and averaged over 1 h including standard deviations denoted as solid gray and light green areas. Thin dashed black and green lines show variances from single-column output (*colX*) at four different grid points determined as departure from a one hour temporal mean. Solid purple lines denote variances from the KIT Doppler lidar including the statistical error according to Lenschow et al. (1994) as error bars, panels a and d, the rotational Raman lidar, panels b and e, and the water vapor differential absorption lidar from University of Hohenheim, panels c and f, (see Tab. 1 for further details). The thin purple (thick light purple) error bars in panels b, c, e and f show the noise (sampling) error according to Lenschow et al. (2000). Gray and light green shaded regions in panels d-f denote the cloud boundaries of PALM and UCLA-LES respectively. See Tab. 3 for the scaling values used.





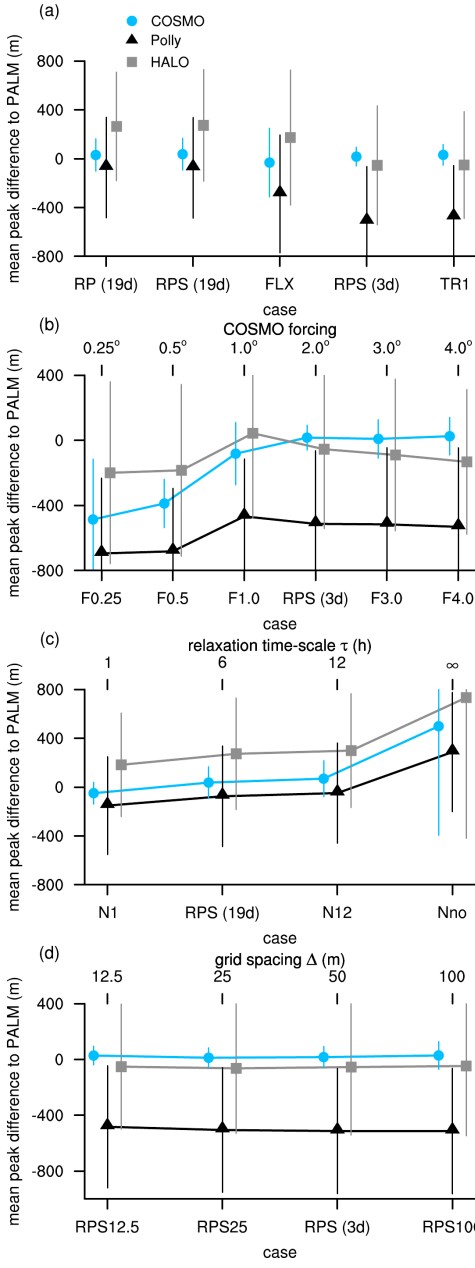

**Figure 8.** Mean peak difference in boundary layer height to PALM between 12 and 14 UTC for the simulated cases listed in Tab. 5. Standard deviations are provided along with the means. Panels b,c and d include the mean peak difference for the sensitivity experiments about the averaging size of the COSMO forcing, the nudging time-scale and the grid spacing, respectively. Panel a lists the remaining cases. Note that the mean peak difference of the PALM reference run on the small domain (*RPS*) is calculated over the whole 19 days (*RPS (19d)*) and the three day testing period (*RPS (3d)*), respectively. The number of values entering the average are (38,147,464) for 19 day runs and (6,38,78) for 3 day runs, respectively. The tuples denote the number of values entering the mean of the difference in boundary layer depth to (COSMO,Polly,HALO).





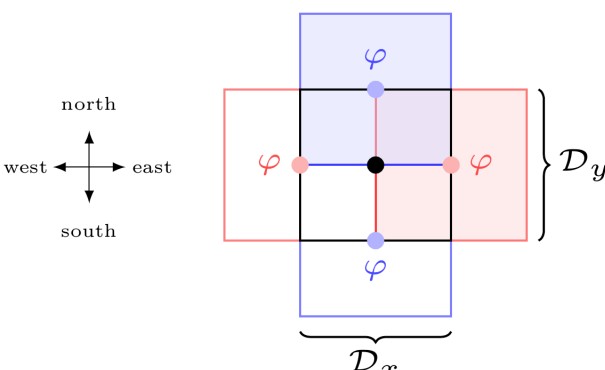

**Figure 9.** Averaging concept for the determination of larger-scale forcing terms from COSMO model output. The shifted domains (red/blue) are used for the calculation of larger-scale gradients of $\phi \in \{\theta_{l,LS}, q_{t,LS}\}$. The centered averaging domain (black) is used for the calculation of all other larger-scale quantities.