# Peer review of "Evaluation of large-eddy simulations forced with mesoscale model output for a multi-week period during a measurement campaign"

_Atmospheric Chemistry and Physics, 2016_

## Referee Comment (RC1) · Anonymous Referee #1 · 13 Jul 2016

This paper describes a set of long ranging LES simulations under realistic conditions, as opposed to the more traditional idealized LES simulations. While the paper does not show an enormous amount of scientific novelty, the long continuous run clearly differentiates itself from previous, similar work done by Neggers or Schalkwijk, and is therefore viable on it's own. Also, it is especially good as a reference for future work by this team and others to have clear description of the methodology, as it is provided in this paper. I can therefore recommend this paper for publication in ACP. However, I would enjoy it if the authors would further the scientific aspects a bit more, for instance in some of the following ways:

*) Presumably, the reason to do these LES simulations in the first place is to retrieve

[Figure]

better quality data than from the COSMO host model. The current results (e.g., fig 6) mainly suggest that the LESs are just close to COSMO. There also seems to be little added value in using 2 different LES codes, at least for the results that are presented here. Does that suggest that the host model strongly dictates the LES result? I understand that it is not the point of this paper to make any of the models (or indeed, observations) look bad, but it would be nice to highlight the benefit of your method by pointing out areas where LES outperforms COSMO, and other areas (such as the nocturnal BL, most likely), where COSMO keeps LES in check.

*) Fig 1: As you mention in the text, the chosen site shows some pretty strong orography and heterogeneity. Is there a way that you could assess the influence of this on your simulations? For instance, a run with interactive land surface with one of the models would be nice.

*) p8, l5: Could this lagging (that you also note later in the paper) be caused by the way the nudging is set up? Since the nudging is always towards the current COSMO state, and with a finite time scale, I'd expect some lag to be present. An experiment with nudging towards a future state (may be an e-folding time away?) could be interesting to test this.

*) p8, l9: Could you quantify the magnitude of the nudging tendencies? For instance with some normalized scatter plots of each of the tendencies for several cases? Figure 2 is not perfect to distinguish what happens for instance during the night, or under specific circumstances. Another way of presenting this could be to look at the aggregate diurnal cycle of the tendencie terms.

*) Table 2: I notice that there are no false positive cumulus days in your simulations. Is this a coincidence, or is this a sign of some bias, for instance in the surface energy balance?

*) p12, l 5: Is there a way you could prove that the advection schemes are to blame for the difference between the LESs? Would the difference for instance be mitigated

at a higher resolution, or is there a way to change the advection scheme in one of the models? A single day run would be sufficient to make the point that numerics matter.

\*) p12, l 21: I'm not entirely convinced about the need for your Richardson based BL depth assessment. I see that this is an attempt to come up with something that applies to all BL types, but then you never discuss the stable boundary layer anyway. Why not stick with something as close as possible to the observed quantities (e.g., max gradient in some profile)? That would make the comparisons you make with those observations a lot more fair, and you could produce a more realistic error/bias assessment.

\*) p18, l 12: I'm not sure why May 5 was chosen as a highlight, at least not as a cumulus day. As the figure shows, it mostly displays forced clouds, with little buoyancy of their own. I would expect this BL to behave like a dry CBL, as it does. If you want to show case a cumulus day, it may be nice to pick another one (if available).

Other comments: \*) p1, l 20: There is quite some newer literature on LES of the stable BL. For instance the GABLS 3 cases (Basu, 2012 or Edwards, BLM 2014) \*) p4, l7: both; momentum <- extra semicolon \*) p5, l20: SUB is not homogeneous because phi=phi(x,y,z), right? so uLS=uLS(z) \*) p6, eq 2: Does this mean that your nudging time scale is constant over height? Have you tried playing with this? \*) p7, l 11: depth of the boundary layer, not height. This may be present a few more times in the manuscript \*) p11, l 31: "Anyhow...." very colloquial. Anyhow's happen a few more times in the manuscript \*) p11, l 31 "the more challenging..." not entirely clear what you mean with this sentence. \*) p13, l9: How does the BL depth perform in SCu topped BLs? I assume that the cloud base is then the desired BL depth, but I can imagine that a strong variability can be observed in those cases. \*) p13, l 21: Why this threshold? It seems arbitrary to me. Can't this be normalized with the maximum variance or so? \*) p15, l9: I printed the paper in Black and White (yes, my department is old fashioned). While I can follow the rest of the manuscript without problems, I could not see the highlighted yellow sections. A better highlight, or just a table with a description of all the days (clear/cumulus/frontal/strong forcing/...) would be nice. \*) p20, l 12: What

does RPS mean? *) p20, l 33: "the question arose" may be a bit more formal? *) P24, l 9: The URL is not yet complete. I assume that will happen later?

---

## Referee Comment (RC2) · Anonymous Referee #2 · 13 Jul 2016

Review of "Evaluation of large-eddy simulations forced with mesoscale model output for a multi-week period during a measurement campaign" by Heinze et al. submitted to ACP.

This manuscript reports analysis of multiday LES simulations of realistic weather phenomena. The forcing comes for a NWP model and results are compared to variety of observations because the simulation period coincides with extensive field observations. I feel this is a very interesting and useful study. It will no doubt lead to many similar follow-up studies where specific aspects will be tested more extensively. The focus in the current manuscript is on the boundary (BL) layer structure (BL height in particular). I find this a little unfortunate as I would like to see more emphasis on simulated clouds as this I feel cloud simulation was at least partly the motivation to go to LES. Perhaps hints offered by Figs. 3 and 4 are sufficient for now.

Overall, I feel this paper should be published after some revisions and clarifications in response to my specific comments below. Some of the comments concern aspects of the models and methodologies beyond the scope of this manuscript, so I do not expect them to be addressed in responses. I do not need to see the revised manuscript.

Specific comments:

1. P1 L20. The key problem with stably stratified BL is the turbulence intermittency and wave-turbulence coupling.

2. P2 L17. I do not think the discussion following the statement "in various single-column and cloud-resolving modeling studies" has anything to do with CRM studies. I suggest revising it to read "in various single-column and LES studies".

3. Section 2.1 and in other places in the text. The terminology applied is not correct. The Boussinesq approximation refers to applying density perturbations only in the gravity term. Thus, most (if not all) nonhydrostatic atmospheric models are Boussinesq (both anelastic and compressible). A more appropriate distinction would be shallow convection approximation (in which case the incompressible assumption is valid and density is constant)) versus deep convection approximation (in which case anelastic or compressible equations are needed). So PALM applies shallow convection approximation (and obviously cannot work properly for deep convection), whereas UCLA-LES is anelastic.

4. A more general comment on model variables (no need to respond, just think about this for the future). Liquid water potential temperature is a perfect variable for shallow nonprecipitating convection. For deep convection, two problems arise. One, the exact formulation is rather cumbersome and I doubt it is used in the two models. Second, when precipitation is considered, then it is not conserved and its sources need to be considered. In contrast, equivalent potential temperature (or moist static energy, MSE) is conserved as long as ice processes are excluded. Should then MSE be used?

5. Another comment on the model. I do not think the water variable is the "total water specific humidity". I think it is (or should be) the mixing ratio. Please do a simple math

exercise with the density of total water, dry air density, and total air density to show that mixing ratio is an appropriate variable when sources (e.g., of water vapor) are considered. The continuity equation for the specific humidity has an extra multiplier for the source term. The equation for the mixing ratio does not, and thus is preferable. Of course in practice the differences are miniscule and can be neglected.

6. P. 4. I think the fundamental differences between Eulerian thermodynamics in UCLA LES and Lagrangian approach in PALM should be better exposed in point 4. These are more significant than the authors realize I think (see below). Also, is PALM really applying saturation adjustment (the paragraph starting on P4L15) and Seifert/Beheng 2-moment microphysics? This is the Lagrangian Cloud Model, so it uses superdroplets, correct?

7. This is perhaps the most significant comment. Formulation of the forcing methodology in section 2.2 comes out of nowhere. I realize that similar methodology has been used by others, but I still think that the methodology can be better explained (perhaps with the exception of the nudging). Please have a look at section 2 in Grabowski et al. (JAS 1996, p. 3684-3709) that formally derived the forcing terms in case of evolving large-scale conditions and later applied it in cloud-resolving model simulations. Perhaps some of the forcing terms derived in that paper are missing in the way LES is forced. Just a thought…

8. The domain shown in Fig. 1 features quite a significant topography. Is then the periodic domain without topography justified? Can one use COSMO output to show if the topography has some effect (e.g., by comparing COSMO results north and south from the observation sites). Does the hill NE to the observation sites affect the observations? Are any observation made from the top of the hill?

9. Section 2.3 does not mention how surface fluxes are calculated. Is the same surface temperature and humidity assumed throughout the LES domain? How variable are surface conditions (e.g., soil type, soil moisture) over the LES domain. Do such considerations matter?

10. I like section 2.4. I think it is important to realize the magnitude of various terms and their evolutions. A small technical comment: the red and orange lines are not easy to distinguish. One of them can be green.

11. Fig. 3 is very nice. It also shows that BL tends to be systematically colder in LES than in observations. Is that a coincidence, or is this true for the entire simulation?

12. P. 12, paragraph below Table 2. I think the origin of the difference between clouds simulated by the two LES models goes beyond just the advection scheme. See comment 6 above. For instance, PALM predicts supersaturation, correct?

13. P13L35. I think this is 5 cm/s, not 50 (this would agree with captions to Fig. 5 and 6).

14. Fig. 6. Despite taking entire page, details of the figure are difficult to see. I suggest breaking the figure into 3 separate figures.

15. The integrated water vapor is commonly referred to as the precipitable water (PW).

16. P20L12-16. I am not sure about the discussion of mesoscale circulations. Since the model excludes topography and (I assume) applies homogeneous surface conditions, the simulated mesoscale circulations may be significantly weaker than in nature.

17. Figure 8 shows that impact of some of the changes is smaller than standard deviations which suggests that the differences may not be statistically significant. How is the standard deviation defined in those plots?

---

## Author Comment (AC1) · 27 Apr 2017

**Authors' response to the comments of Referee #I on the manuscript**

*Evaluation of large-eddy simulations forced with mesoscale output for a multi-week period during a measurement campaign*

by R. Heinze et al.

submitted to ACP

27[th] April, 2017

Dear Referee I,

Thank you very much for your comments on the manuscript. In what follows we present our replies (plain text) to your comments and questions (*italics*). The line numbers and the page numbers refer to the revised manuscript (without mark-up) which is attached to this response.

- *\*) Presumably, the reason to do these LES simulations in the first place is to retrieve better quality data than from the COSMO host model. The current results (e.g., fig 6) mainly suggest that the LESs are just close to COSMO. There also seems to be little added value in using 2 different LES codes, at least for the results that are presented here. Does that suggest that the host model strongly dictates the LES result? I understand that it is not the point of this paper to make any of the models (or indeed, observations) look bad, but it would be nice to highlight the benefit of your method by pointing out areas where LES outperforms COSMO, and other areas (such as the nocturnal BL, most likely), where COSMO keeps LES in check.*

  The primary reason to do these simulations is to receive turbulence-resolving modeling output localized at a special region with a fairly simple (semi-idealized) setup. It is highly desirable to get added value from the LES in variables that are determined by small-scale processes where turbulence plays a role, such as higher-order moments or microphysics. It is true that Fig. 6 (now Figs. 6-8) shows that the LES output is just close to COSMO which shows that the larger-scale forcing has a decent impact on the results. Anyhow, it should be kept in mind that COSMO is an operational NWP-model highly tuned compared to LES models running in a semi-idealized setup where only warm microphysics are used. A comment on the closeness of the LES to COSMO was made in the conclusion section (p. 22, l. 32). Another point where the LES are slightly better than COSMO is the specific rain water. COSMO shows much more rain than observed (p. 12, l. 21).

  Looking closer at Fig. 2, it can be seen that larger-scale forcing and nudging is as important as the fast LES physics during night-times where our 50m-LES is much too coarse to resolve the stable boundary

layer. Thus, here COSMO keeps the LES in check as the referee proposed. However, this should not affect the daytime development as previous work (van Stratum and Stevens, 2015) showed that the influence of biases in the representation of the nocturnal boundary layer do not substantially influence the subsequent daytime development. A small paragraph was added to section 2.4 (p. 8, l. 19-22).

Furthermore, on days with strong impact on the larger-scales, also the larger-scale tendencies dominate slaving the LES output to COSMO by construction of the forcing. A real benefit from LES is that it can provide variance (turbulence) profiles that are in good agreement with lidar observations (see Fig. 9). Unfortunately, the variances are not available from the COSMO-output so that a direct comparison to COSMO is not possible at this stage.

To conclude, the LES has the advantage in representing the structure of clouds and microphyscial changes in shallow convection and its coupling to the boundary layer. Still, in the way the LES is forced it is also in someways held back by COSMO. Thus, where there are improvements they are modest.

- *) Fig 1: As you mention in the text, the chosen site shows some pretty strong orography and heterogeneity. Is there a way that you could assess the influence of this on your simulations? For instance, a run with interactive land surface with one of the models would be nice.*

  Considering the influence of the orography and the heterogeneous surface would require some major changes in the setup. As we are using horizontal cyclic boundary conditions, a sufficient buffer zone around the analysis region would be needed to overcome the influence of the cyclicity of the flow. Thus, the modeling domain would have to become larger in dependence on wind-speed and wind-direction. Considerably more effort would be needed to study the influence of the real-world heterogeneities on the flow.

  A run with an interactive land surface model instead of prescribing surface temperature and humidity could be a first step towards assessing the influence of a more realistic surface. But still, the land surface model as it is implemented in PALM and UCLA-LES needs input like soil and vegetation type which has to be prescribed horizontally homogeneous. In general we refrained from using the land surface model in the two LES models to facilitate the comparison between the two models.

  Nonetheless, there have been comparisons between simulations around Jülich in a semi-idealized setup with PALM, UCLA-LES and the Dutch Atmospheric Large-Eddy Simulation model DALES. In the simulations with DALES, a land-surface model based on the IFS-scheme was used. The following figures show comparison of the models with observations for a 3-day run (24-26 April 2013). Note that the DALES simulation differs from the PALM and UCLA simulations also in other aspects (different radiation scheme and large-scale forcing data set) which prohibits to assess how large the influence of

the land-surface scheme is.

[Figure]

DALES is able to reproduce the surface sensible heat flux much better than PALM and UCLA-LES. The surface latent heat flux is overestimated strongly on 24 and 25 April and is much higher than in the simulations with PALM and UCLA-LES. The boundary layer depth in DALES is comparable to PALM and UCLA-LES. Thus, even though an interactive land-surface model is used (and radiation and another forcing data set) in DALES, the difference to the observations cannot be significantly reduced.

To asses the influence of surface heterogeneity and orography around Jülich the limited-area LES over Germany presented in Heinze et al. (2017) can be used (see also the statement on p. 23 l. ??). The present manuscript can be seen as basis for a further study were we try to understand which importance the mesoscale forcing coming from orography and surface heterogeneity has. In this ongoing work comparisons are made between the simulations with ICON in the semi-idealized setup, limited-area LES with ICON on smaller domains centered around Jülich and the limited-area LES over Germany.

Please also see our answer to point 8 of referee #II (p. 3).

- *) p8, l5: Could this lagging (that you also note later in the paper) be caused by the way the nudging is set up? Since the nudging is always towards the current COSMO state, and with a finite time scale, I'd expect some lag to be present. An experiment with nudging towards a future state (may be an e-folding time away?) could be interesting to test this.*

We agree that the way the nudging is set up, a lag is to be expected. The following figure shows how the boundary layer depth evolves in case that different nudging time-scales are used. The first 5 days of the simulation-period are shown. REF denotes the simulation with a nudging time-scale of 6 h, N1 has a time-scale of 1 h and N12 a time-scale of 12 h. No considerable differences occur between REF and N12. In run N1, however, $z_i$ agrees more with COSMO than with REF, which can be clearly observed for Days 4 and 5 and which indicates a large impact of the nudging terms in this case. The stronger (tighter) the nudging the smaller the lag.

[Figure]

Nudging towards a future state is a very interesting idea. Unfortunately, it would go beyond the scope of the present paper.

- *) p8, l9: Could you quantify the magnitude of the nudging tendencies? For instance with some normalized scatter plots of each of the tendencies for several cases? Figure 2 is not perfect to distinguish what happens for instance during the night, or under specific circumstances. Another way of presenting this could be to look at the aggregate diurnal cycle of the tendency terms.*

We followed your suggestion and calculated the aggregated diurnal cycle of the tendency terms by averaging the tendency terms shown in Fig. 2 over the 19 days. The results are provided in the following figure.

[Figure]

On average the fast physics (LES tendencies) are of the same order of magnitude as larger-scale advection (LSA), subsidence (SUB) and nudging (NUD) during the night. For the liquid water potential temperature budget, advection and nudging lead both to cooling between 0h and about 9h. The effect of nudging on the total water specific humidity budget is rather small between 20h and 7h. During this time, larger-scale horizontal and vertical advection act simultaneously to dry the atmosphere.

- *) Table 2: I notice that there are no false positive cumulus days in your simulations. Is this a coincidence, or is this a sign of some bias, for instance in the surface energy balance?*

This is a good point. A false positive cumulus day means that shallow clouds were simulated although they have not been observed. There are indeed no such days in our simulations. Looking also at Fig. 6b we can see that there are several days where the peak surface latent heat flux in the LES is lower than the observed weighted average (e.g., 27-30 April, 7-12 May). This could hint to a dry bias where the simulated boundary layer is not moist enough to form capping clouds.

- *) p12, l 5: Is there a way you could prove that the advection schemes are to blame for the difference between the LESs? Would the difference for instance be mitigated at a higher resolution, or is there a way to change the advection scheme in one of the models? A single day run would be sufficient to make the point that numerics matter.*

We did an additional simulation with PALM on the small horizontal domain (setup RPS, see Tab. 5) where the advection scheme for momentum and scalars was changed from a fifth-order scheme based on Wicker and Skamarock (2002) [WS] to a second-order scheme based on Piacsek and Willimas (1970) [PW]. The following figure shows cloud and rain water in the lower 5 km of the modeling domain for WS (upper figure) and PW (lower figure). The same contours as in Fig. 4 are used.

[Figure]

It can be seen that the cloud water is slightly sensitive to the advection scheme. On days with capping cloud layers (28/4, 29/4, 3/5), the WS-scheme produces slightly larger qc than the PW-scheme. The sensitivity of rain water is stronger leading to higher rain water on 7/5, 11/5 and 12/5 in case the PW-scheme is used. Overall, the influence of the advection scheme is small but numerics do mater nonetheless.

- *) p12, l 21: I'm not entirely convinced about the need for your Richardson based BL depth assessment. I see that this is an attempt to come up with something that applies to all BL types, but then you never discuss the stable boundary layer anyway. Why not stick with something as close as possible to the observed quantities (e.g., max gradient in some profile)? That would make the comparisons you make with those observations a lot more fair, and you could produce a more realistic error/bias assessment.*

We decided to use a more universal criterion like the height of a critical Richardson number after carefully comparing several methods. The following figure shows a comparison between the chosen method (bulk Richardson number, black), the vertical location of the average minimum sensible heat flux (standard flux method, blue) and the vertical location of the largest increase of potential temperature (gradient method, red) for the simulation of 5 days (24-28 April 2013).

[Figure]

It can be seen that the standard flux and th gradient methods do not yield satisfying solutions for this simulation. They show unrealistic sudden jumps of $z_i$ and fail at night-time (i.e., no boundary layer height is diagnosed at all). Furthermore, the bulk Richardson number method, however, produces a relatively smooth solution and is able to determine a $z_i > 0$ m at all times for this simulation. During the daytime of Days 1 and 2 typical CBLs form, and general agreement of the results of all three methods is found. To a lesser extend, this also applies for Day 5. During the frontal passage on Day 3 and the following post-frontal situation on Day 4, the flux and gradient method yield no trustworthy solutions.

- *) p18, l 12: I'm not sure why May 5 was chosen as a highlight, at least not as a cumulus day. As the figure shows, it mostly displays forced clouds, with little buoyancy of their own. I would expect this BL to behave like a dry CBL, as it does. If you want to show case a cumulus day, it may be nice to pick another one (if available).*

You are absolutely right, May 5 is not the perfect show-case for a shallow cumulus topped boundary layers. Unfortunately, other days with a prototype shallow cumulus layer are not available.

- *Other comments:*
  - *) p1, l 20: There is quite some newer literature on LES of the stable BL. For instance the GABLS 3 cases (Basu, 2012 or Edwards, BLM 2014)*

    A reference to Edwards et al. (2014) and Ansorge and Mellado (2014, 2016) was added (p. 2, l. 1)

  - *) p4, l7: both; momentum <- extra semicolon*

    Comma was replaced by semicolon (p. 4, l. 15)

  - *) p5, l20: SUB is not homogeneous because phi=phi(x,y,z), right? so uLS=uLS(z)*

It is correct that the SUB tendency is not homogeneous which is already reflected in the text (p 5, l. 32)

- *) p6, eq 2: Does this mean that your nudging time scale is constant over height? Have you tried playing with this?*

Correct, the nudging time scale is constant over height. We played with the profile of the nudging time-scale. The following figures show the tested profiles and the effect on the boundary layer depth.

[Figure]

[Figure]

For the Nudging Profile 1, the nudging time scale is set to $10^{10}$ s between the surface and 0.5 $z_i$ (which results in having no nudging at all). Between 0.5 $z_i$ and $z_i$, the time scale is then brought to 6 h via a cosine-function. For profile 2 and 3 the same procedure is applied between 0.5 $z_i$ and $z_i$ (Profile 2), and $z_i$ and 2 $z_i$ (Profile 3), respectively. Test simulations of 24 April 2013 (IOP 6) show that in terms of the boundary layer depth, the effect is minor during the convective hours (10 UTC to 16 UTC). All realizations lie in the spread of the measurements.

○ *) p7, l 11: depth of the boundary layer, not height. This may be present a few more times in the manuscript

The word *height* was replaced by *depth* where appropriate.

○ *) p11, l 31: "Anyhow...." very colloquial. Anyhow's happen a few more times in the manuscript

The word *anyhow* was replaced by *however* where appropriate.

○ *) p11, l 31 "the more challenging..." not entirely clear what you mean with this sentence.

We wanted to make the point that simulating proper shallow cumulus layers on 25 April and 1 May is difficult for the LES. The sentence was rephrased to reflect this (p. 12, l. 12).

○ *) p13, l9: How does the BL depth perform in SCu topped BLs? I assume that the cloud base is then the desired BL depth, but I can imagine that a strong variability can be observed in those cases.

In the 19-day period which was simulated there was unfortunately no real stratocumulus-day. The cloud structure on 10 May resembled stratocumulus as close as possible around noon. In Fig. 3l (p. 32), it can be seen that the cloud top in PALM is around 1.3 km (maximum of qc) which is comparable to the value of $z_i$ during noon (see Fig. 5c, p. 34). Thus, cloud base and $z_i$ based on the bulk Richardson number criterion resemble.

- *) p13, l 21: Why this threshold? It seems arbitrary to me. Can't this be normalized with the maximum variance or so?*

  In Schween et al. (2014) a detailed discussion on the choice of the threshold and its sensitivity to the derived mixing layer height can be found.

- *) p15, l9: I printed the paper in Black and White (yes, my department is old fashioned). While I can follow the rest of the manuscript without problems, I could not see the highlighted yellow sections. A better highlight, or just a table with a description of all the days (clear/cumulus/frontal/strong forcing/...) would be nice.*

  We changed the highlighting from light yellow to black stippled in Figs. 5-8.

- *) p20, l 12: What does RPS mean?*

  RPS is the name of a case listed in Table 5 (p. 20), S denotes the usage of a small horizontal modeling domain. A hint is given in the text now. (p 20, l. 6)

- *) p20, l 33: "the question arose" may be a bit more formal?*

  The sentence was changed to "the question came up ..." . (p 21, l. 7)

- *) P24,l 9: The URL is not yet complete. I assume that will happen later?*

  Correct.

[revised manuscript text omitted]
| LWP | liquid water path | HATPRO | | | Steinke et al. (2015) |
| shf, | surface sensible heat flux, | energy balance stations | KITcube | 04/24-05/12 | Kalthoff et al. (2013), |
| lhf | surface latent heat flux | | Kc Wasserwerk | | Maurer et al. (2016), |
| | | | TER Selhausen | | Graf et al. (2010), |
| | | | TER Niederzier | | Zacharias et al. (2011) |
| | | | TER Ruraue | | |
| $\overline{w'^2}$ | vertical velocity variance | Doppler lidar WLS7-V2 | KITcube | 04/24, 05/05 | Maurer et al. (2016) |
| | | ($z < 400$ m), Doppler | | | |
| | | lidar WindTracer | | | |
| | | WTX ($z \geq 400$ m) | | | |
| $\overline{T'^2}$ | temperature variance | rotational Raman lidar RRL | KITcube | 04/24, 05/05 | Behrendt et al. (2015) |
| $\overline{\rho_v'^2}$ | absolute humidity variance | water vapor differential | KITcube | 04/24, 05/05 | Muppa et al. (2016) |
| | | absorption lidar WVDIAL | | | |
| $z_{cb}$, | cloud base height, | Cloudnet | JOYCE | 04/24-05/12 | Illingworth et al. (2007) |
| $z_{ct}$, | cloud top height, | Cloudnet | LACROS | | Illingworth et al. (2007) |
| $d_c$ | cloud layer depth | ceilometer CHM15k | JOYCE | | Löhnert et al. (2015) |

[revised manuscript text omitted]

---

## Author Comment (AC2) · 27 Apr 2017

**Authors' response to the comments of Referee #II on the manuscript**

*Evaluation of large-eddy simulations forced with mesoscale output for a multi-week period during a measurement campaign*

by R. Heinze et al.

submitted to ACP

27[th] April, 2017

Dear Referee II,

Thank you very much for your comments on the manuscript. In what follows we present our replies (plain text) to your comments and questions (*italics*). The line numbers and the page numbers refer to the revised manuscript (without mark-up) which is attached to this response.

- *1.P1 L20. The key problem with stably stratified BL is the turbulence intermittency and wave-turbulence coupling.*

  This information was added (p. 1, l. 20).

- *2. P2 L17. I do not think the discussion following the statement "in various single-column and cloud-resolving modeling studies "has anything to do with CRM studies. I suggest revising it to read "in various single-column and LES studies".*

  In this sentence, "cloud resolving modeling" was replaced by "LES" (p. 2, l. 18).

- *3. Section 2.1 and in other places in the text. The terminology applied is not correct. The Boussinesq approximation refers to applying density perturbations only in the gravity term. Thus, most (if not all) nonhydrostatic atmospheric models are Boussinesq (both anelastic and compressible). A more appropriate distinction would be shallow convection approximation (in which case the incompressible assumption is valid and density is constant)) versus deep convection approximation (in which case anelastic or compressible equations are needed). So PALM applies shallow convection approximation (and obviously cannot work properly for deep convection), whereas UCLA-LES is anelastic.*

  The terminology "Boussinesq approximation" was replaced by "shallow-convection approximation" (p. 4, l. 10 and p. 6, l. 29).

- *4. A more general comment on model variables (no need to respond, just think about this for the future).*

*Liquid water potential temperature is a perfect variable for shallow nonprecipitating convection. For deep convection, two problems arise. One, the exact formulation is rather cumbersome and I doubt it is used in the two models. Second, when precipitation is considered, then it is not conserved and its sources need to be considered. In contrast, equivalent potential temperature (or moist static energy, MSE) is conserved as long as ice processes are excluded. Should then MSE be used?*

Thanks for pointing this out. There is also a problem with MSE as it is not conserved for isobaric ascent, i.e., when isobars are not aligned with iso-heights. It also has difficulties with the ice phase and is only approximately conserved in the face of precipitation.

- *5. Another comment on the model. I do not think the water variable is the "total water specific humidity". I think it is (or should be) the mixing ratio. Please do a simple math exercise with the density of total water, dry air density, and total air density to show that mixing ratio is an appropriate variable when sources (e.g., of water vapor) are considered. The continuity equation for the specific humidity has an extra multiplier for the source term. The equation for the mixing ratio does not, and thus is preferable. Of course in practice the differences are minuscule and can be neglected.*

Indeed, your are correct. Both models use mixing ratios as mass fraction and not specific humidities. The text was changed accordingly (e.g., p. 4, l. 24).

- *6. P. 4. I think the fundamental differences between Eulerian thermodynamics in UCLA LES and Lagrangian approach in PALM should be better exposed in point 4. These are more significant than the authors realize I think (see below). Also, is PALM really applying saturation adjustment (the paragraph starting on P4L15) and Seifert/Beheng 2-moment microphysics? This is the Lagrangian Cloud Model, so it uses superdroplets, correct?*

Point 4 on page 4 was thought to show some different fields of study in which PALM and UCLA-LES are generally used. Whereas PALM is well known to have a Lagrangian cloud model focusing on shallow convection, UCLA-LES is widely used to further develop and test microphysical schemes in an Eulerian approach. Point 4 was not meant to reflect specifically what was used in the present simulations. As described from page 4, line 24 onwards, PALM and UCLA-LES both used saturation adjustment and the warm 2-moment microphysics scheme. Thus, PALM does not make use of the Lagrangian cloud model in the present simulations.

The text was changed to point out more strongly that both models use saturation adjustment in the present simulations (p. 4, l. 25-26).

- *7. This is perhaps the most significant comment. Formulation of the forcing methodology in section 2.2 comes out of nowhere. I realize that similar methodology has been used by others, but I still think that the methodology can be better explained (perhaps with the exception of the nudging). Please have a look at section 2 in Grabowski et al. (JAS 1996, p. 3684-3709) that formally derived the forcing terms in case of evolving large-scale conditions and later applied it in cloud-resolving model simulations. Perhaps some of the forcing terms derived in that paper are missing in the way LES is forced. Just a thought...*

We agree that the methodology on how the larger-scale forcing components are derived is missing. We added a small paragraph to point out that the forcing terms have a sound physical basis and that they are derived in detail for example in Grabowski et al. (1996) (p. 5, l. 16-19). We refrain from presenting the methodology anew as Grabowski et al. (1996) give a precise description on how to derive the larger-scale forcing terms.

- *8. The domain shown in Fig. 1 features quite a significant topography. Is then the periodic domain without topography justified? Can one use COSMO output to show if the topography has some effect (e.g., by comparing COSMO results north and south from the observation sites). Does the hill NE to the observation sites affect the observations? Are any observation made from the top of the hill?*

Considering the influence of the topography would require some major changes in the present setup. As we are using horizontal cyclic boundary conditions, a sufficient buffer zone around the analysis region would be needed to overcome the influence of the cyclicity of the flow. Thus, the modeling domain would have to become larger in dependence on the wind-speed and wind-direction. Considerably more effort would be needed to study the influence of the real-world topography on the flow.

The COSMO output which is used here for driving the LES cannot be used to study the effect of the hill as the output is already horizontally averaged over squares where Jülich is in the centre (see Fig. 11 (p. 40)). Thus, the original COSMO fields would have to be obtained from the archive of the German Weather Service to answer this question.

The measurements during HOPE were taken between the debris hills and unfortunately not on top of the hill Sophienhöhe (see also the ACP HOPE overview paper of Macke et al. (2017)). Figure 1b shows the three principal measurement sites which are located in the rather flat region between the two debris hills. Thus, we cannot assess the influence of the hill on the measurements.

To asses the influence of surface heterogeneity and orography around Jülich the limited-area LES over Germany presented in Heinze et al. (2017) can be used (see also the statement on p. 23 l. 30). The present manuscript can be seen as basis for a further study were we try to understand which importance

the mesoscale forcing coming from orography and surface heterogeneity has. In this ongoing work comparisons are made between the simulations with ICON in the semi-idealized setup, limited-area LES with ICON on smaller domains centered around Jülich and the limited-area LES over Germany.

Please also see our answer to point 2 of referee #I (p. 2).

- *9. Section 2.3 does not mention how surface fluxes are calculated. Is the same surface temperature and humidity assumed throughout the LES domain? How variable are surface conditions (e.g., soil type, soil moisture) over the LES domain. Do such considerations matter?*

At the surface, temperature and humidity are prescribed horizontally homogeneous (see p. 7, l. 8). Then Monin-Obukhov similarity theory is used to calculate surface sensible and latent heat flux (see p. 14, l. 23). A statement was added in section 2.3 to point out how the surface fluxes are derived (p 7, l. 10).

The surface conditions are horizontally homogeneous but change in time as surface temperature and humidity are taken from the larger-scale forcing data set. In the LES models, no land-surface model is used to facilitate the comparison between the two LES models. Thus, parameters like soil type, vegetation type or soil moisture do no enter our simulations.

- *10. I like section 2.4. I think it is important to realize the magnitude of various terms and their evolutions. A small technical comment: the red and orange lines are not easy to distinguish. One of them can be green.*

The color line of the nudging tendency in Fig. 2 was changed from orange to violet. We refrain from using red and green in one figure as red-green color blindness is rather widespread.

- *11. Fig. 3 is very nice. It also shows that BL tends to be systematically colder in LES than in observations. Is that a coincidence, or is this true for the entire simulation?*

Looking at Fig. 7c where the potential temperature at a height of 25 m is shown, it can be seen that the LES are usually colder during day times (with some exceptions on 26 April, 27 April,11 May) in the boundary layer. Thus, there seems to be a tendency that the simulations are a bit too cold during daytimes compared to radio soundings. This information was added in the manuscript (p. 15, l. 12).

- *12. P. 12, paragraph below Table 2. I think the origin of the difference between clouds simulated by the two LES models goes beyond just the advection scheme. See comment 6 above. For instance, PALM predicts supersaturation, correct?*

As already stated in the answer to comment 6, the Lagrangian cloud model was not used in PALM. Thus, PALM and UCLA-LES both use saturation-adjustment and the same microphysics scheme. Additional simulations with another advection scheme in PALM are presented in the answer to referee I and show that cloud and rain water are slightly dependent on the advection scheme.

- *13. P13L35. I think this is 5 cm/s, not 50 (this would agree with captions to Fig. 5 and 6).*

The typo was corrected (p. 14, l. 12).

- *14. Fig. 6. Despite taking entire page, details of the figure are difficult to see. I suggest breaking the figure into 3 separate figures.*

Figure 6 was broken into 3 separate figures (Fig. 6, 7, 8).

- *15. The integrated water vapor is commonly referred to as the precipitable water (PW).*

We decided to retain the formulation "integrated water vapor" as it can synonymously be used with precipitable water.

- *16. P20L12-16. I am not sure about the discussion of mesoscale circulations. Since the model excludes topography and (I assume) applies homogeneous surface conditions, the simulated mesoscale circulations may be significantly weaker than in nature.*

We totally agree that the simulated mesoscale circulation may be weaker than in nature as orography and surface heterogeneities are neglected. We changed the description by naming these circulations internally generated  (p. 19 l. 32; p. 23 l. 7 ). Furthermore, the semi-idealized simulations presented in the manuscript can be seen as first step towards assessing the role of the mesoscale as there is ongoing work to explore the effect of orography and surface heterogeneity by comparing simulations with ICON in semi-idealized setup and limited-area setup (p 3., l. 3).

- *17. Figure 8 shows that impact of some of the changes is smaller than standard deviations which suggests that the differences may not be statistically significant. How is the standard deviation defined in those plots?*

The metric *mean peak difference to PALM* including the standard deviation is calculated as follows and possibly best followed while also looking at Fig. 5 where the temporal evolution of the boundary layer depth of different sources are shown. For the metric, the absolute difference PALM – COSMO  or observation is taken for every available data point. (We refrained from using the norm of the difference

in order to obtain an unbiased result.) Then only those values are further processed which occur between 12 and 14 UTC for each day.  This data subset is then averaged and the standard deviation is calculated as the square root of the variance of the data subset.

The large standard deviation in some cases can be attributed to the large scatter in the observations. Figure 5 shows that for example on 24 April Polly and Halo are close to PALM whereas they deviate noticeably on 25 and 26 April. Thus, a large scatter in the peak difference to PALM can be expected. As for some sensitivity studies only 3 days instead of 19 days are simulated (see Tab. 5) the number of values entering the statistics is also quite small in comparison to the statistics over 19 day runs.

[revised manuscript text omitted]

---

## Author Comment (AC3) · 27 Apr 2017

Dear Referee,

find attached to this post the updated manuscript, including the markup. The final manuscript file without markup was already attached to the reply letter.

Please also note the supplement to this comment:
http://www.atmos-chem-phys-discuss.net/acp-2016-498/acp-2016-498-AC3-supplement.pdf